# A Theoretical and Empirical Study on the Convergence of Adam with an *"Exact"* Constant Step Size in Non-Convex Settings

## Abstract

In neural network training, RMSProp and Adam are widely favoured optimization algorithms. A key factor in their performance is selecting the correct step size, which can greatly influence their effectiveness. Moreover, the theoretical convergence properties of these algorithms remain a subject of significant interest. This article provides a theoretical analysis of a constant step size version of Adam in non-convex settings. It discusses the importance of using a fixed step size for Adam's convergence. We derive a constant step size for Adam and offer insights into its convergence in non-convex optimization scenarios. Firstly, we show that deterministic Adam can be affected by rapidly decaying learning rates, such as linear and exponential decay, which are often used to establish tight convergence bounds for Adam. This suggests that these rapidly decaying rates play a crucial role in driving convergence. Building on this observation, we derive a constant step size that depends on the dynamics of the network and the data, ensuring that Adam can reach critical points for smooth and non-convex objectives, with provided bounds on running time. Both deterministic and stochastic versions of Adam are analyzed, and we establish sufficient conditions for the derived constant step size to achieve asymptotic convergence of the gradients to zero with minimal assumptions. We conduct experiments to empirically compare Adam's convergence with our proposed constant step size against state-of-the-art step size schedulers on classification tasks. Furthermore, we demonstrate that our derived constant step size outperforms various state-of-the-art learning rate schedulers and a range of constant step sizes in reducing gradient norms. Our empirical results also indicate that, despite accumulating a few past gradients, the key driver for convergence in Adam is the use of non-increasing step sizes.

## 1 Introduction

Optimisation problems in machine learning are often structured as minimizing a finite sum $\min_{x \in \mathbb{R}^n} f(x)$, where $f : \mathbb{R}^n \mapsto \mathbb{R}$ and $f(x) = \frac{1}{k} \sum_{i=1}^{k} f_i(x)$. Each $f_i : \mathbb{R}^n \mapsto \mathbb{R}$ typically exhibits non-convex behaviour, particularly in neural network domains. A prominent method for tackling such problems is *Stochastic Gradient Descent* (SGD), where updates occur iteratively as $x_{t+1} := x_t - \alpha \nabla \tilde{f}_{i_t}(x_t)$, with $\alpha$ denoting the step size and $\tilde{f}_{i_t}$ being a randomly chosen function from $\{f_1, f_2, ..., f_k\}$ at each iteration $t$. SGD is preferred for training deep neural networks due to its computational efficiency, especially with mini-batch training, even with large datasets (Bottou, 2012).

Adaptive variants of SGD, which incorporate past gradients through averaging, have gained traction due to their ability to track gradient scaling on an individual parameter basis (Bottou, 2009). These methods are preferred for their perceived ease of tuning compared to traditional SGD. Adaptive gradient methods typically update using a vector obtained by applying a linear transformation, often referred to as *diagonal pre-conditioner* to the linear combination of all previously observed gradients. This pre-conditioning is believed to enhance algorithm robustness to hyperparameter choices, making them less sensitive to initial settings.

Adagrad (Duchi et al., 2011; McMahan & Streeter, 2010) showed superior performance, especially in scenarios with small or sparse gradients. However, its effectiveness diminishes in situations with non-convex loss functions and dense gradients due to rapid learning rate decay. To address this, variants such as RMSProp (Tieleman & Hinton, 2012), Adam (Kingma & Ba, 2015) (**Algorithm 1**), Adadelta (Zeiler, 2012), and Nadam (Dozat, 2016) have been proposed. These methods mitigate learning rate decay by employing exponential moving averages of squared past gradients, thereby limiting the reliance on all past gradients during updates. While these algorithms have found success in various applications (Vaswani et al., 2019), they have also been observed to exhibit non-convergence in many settings (Sashank et al., 2018), especially in deterministic environments where noise levels are controlled during optimisation. This regulation is accomplished either by employing larger batches (Martens & Grosse, 2015; De et al., 2017; Babanezhad Harikandeh et al., 2015) or by integrating variance-reducing techniques (Johnson & Zhang, 2013; Defazio et al., 2014).

For the reader's convenience, we first clarify a few necessary notations used in the forthcoming Generic Adam. First, we denote $\mathbf{w} \in \mathbb{R}^d$ as the optimising variable and $\mathbf{g}_t$ as the stochastic gradient at the $t^{th}$ iteration respectively, $\alpha > 0$ base learning rate, $0 < \rho << 1$ as the stabilizer, and $\beta_1, \beta_2 \in (0, 1]$ momentum parameter, respectively. Denote $\mathbf{0} = (0, \ldots, 0)^T \in \mathbb{R}^d$, and $\mathbf{1} = (1, \ldots, 1)^T \in \mathbb{R}^d$. All operations, such as multiplying, dividing, and taking the square root, are executed in the coordinate-wise mode.

---

**Algorithm 1:** Adam Algorithm Pseudocode (Kingma & Ba, 2015)

---

**Input:** Learning rate: $\alpha \in (0, 1]$, $\beta_1, \beta_2 \in [0, 1)$, initial starting point $\mathbf{w}_0 \in \mathbb{R}^d$, a constant vector
    $\rho \mathbf{1} > \mathbf{0} \in \mathbb{R}^d$, we assume we have access to a noisy oracle for gradients of $f : \mathbb{R}^d \mapsto \mathbb{R}$

**1 Initialization: $\mathbf{m}_0 = \mathbf{0}$, $\mathbf{v}_0 = \mathbf{0}$**
**2 for** $t$ *from 1 to $T$:* **do**
**3**     $\mathbf{g}_t = \nabla f(\mathbf{w}_t)$
**4**     $\mathbf{m}_t = \beta_1 \mathbf{m}_{t-1} + (1 - \beta_1)\mathbf{g}_t$
**5**     $\mathbf{v}_t = \beta_2 \mathbf{v}_{t-1} + (1 - \beta_2)\mathbf{g}_t^2$
**6**     $\mathbf{V}_t = \mathtt{diag}(\mathbf{v}_t)$
**7**     $\mathbf{w}_{t+1} = \mathbf{w}_t - \alpha(\mathbf{V}_t^{1/2} + \mathtt{diag}\,(\rho\mathbf{1}))^{-1}\,\mathbf{m}_t$
**8 End**

---

Despite their widespread adoption in solving neural network problems, adaptive gradient methods like RMSProp and Adam often lack theoretical justifications in non-convex scenarios, even when dealing with exact or deterministic gradients (Bernstein et al., 2018). Several sufficient conditions have been proposed to guarantee the global convergence of Adam, which can be further classified into the following two categories:

**(B1) Learning rate decay**: The main cause of divergences in Adam and RMSProp predominantly arises from the disparity between the two successive learning rates (Sashank et al., 2018).

$$\Gamma_t = \frac{\sqrt{\mathbf{V}_t}}{\alpha_t} - \frac{\sqrt{\mathbf{V}_{t-1}}}{\alpha_{t-1}} \tag{1}$$

If the positive definiteness property of $\Gamma_t$ is violated, Adam and RMSProp may experience divergence. The $\Gamma_t > 0$ constraint can be relaxed (Barakat & Bianchi, 2020), and convergence of Adam can still be achieved when the condition $\frac{\sqrt{\mathbf{V}_t}}{\alpha_t} \geq c\frac{\sqrt{\mathbf{V}_{t-1}}}{\alpha_{t-1}}$ holds for all $t$ and some positive constant $c$. Prior research efforts (Shi et al., 2020; Tian & Parikh, 2022; Chen et al., 2018; Luo et al., 2019; Défossez et al., 2020; Zou et al., 2019) have focused on establishing convergence properties of optimisation algorithms such as RMSProp and Adam. These investigations typically utilize step size schedules of the form $\alpha_t = \frac{\alpha}{t^a}$ for all $t \in \{1, 2, \ldots, T\}$, $\alpha \in (0, 1)$, and $a > 0$, or other time-dependent variations like $\alpha_t = \alpha(1 - \beta_1)\sqrt{\frac{1-\beta_2^t}{1-\beta_2}}$. While a non-increasing (diminishing) step size is crucial for convergence, empirical analysis suggests that rapid decay of the step size can lead to sub-optimal outcomes, as discussed in the next section.

**(B2) Temporal decorrelation**: The divergence observed in RMSProp stems from imbalanced learning rates (Zhou et al., 2018; Sashank et al., 2018) rather than the absence of $\Gamma_t > 0$. Leveraging this insight,

AdaShift (Zhou et al., 2018) was proposed, which incorporates a temporal decorrelation technique to address the inappropriate correlation between $\mathbf{v}_t$ and the current second-order moment $\mathbf{g}_t^2$. This approach requires the adaptive learning rate $\alpha_t$ to be independent of $\mathbf{g}_t^2$. However, it's worth noting that the convergence analysis of AdaShift was primarily limited to RMSProp for resolving the convex counterexample presented in (Sashank et al., 2018).

In contrast to the aforementioned modifications and restrictions, we introduce an alternative *sufficient condition* (abbreviated as **SC**) to ensure the global convergence of the original Adam. The proposed **SC** (refer to **Section** 4.1) depends on a constant learning rate $\alpha$ [our proposed learning rate] and on the parameter $\beta_1$. Our **SC** doesn't necessitate rapidly decaying step sizes and positive definiteness of $\Gamma_t$. It's easier to verify and more practical compared to (**B2**).

**NOTE**: A recent study (Chen et al., 2022) proposes a similar convergence rate to ours using a constant step size approach for mini-batch Adam, where they employ $\alpha_t = \frac{\alpha}{\sqrt{T}}$, with $\alpha$ being any positive real number. However, our work asks the questions:

1. For better convergence[1] of Adam with non-decreasing or constant step sizes, is $\alpha_t = \frac{\alpha}{\sqrt{T}}$ sufficient?

2. Are there any other terms related to the dynamics of the model and data that need to be integrated as constants in the learning rates proposed above?

3. If such terms exist, will they lead to better convergence?

Our work specifies the *exact* learning rate that ensures convergence with fewer assumptions and contains information about the model dynamics and data distribution in the form of *Lipschitz smoothness of the loss*.

*This article is organized as follows:* **Section** 2 presents basic definitions and pseudocodes that will be used throughout our theoretical analysis. In **Section** 3, we provide the motivation for this work and outline the major contributions of the article. **Section** 4 discusses the convergence guarantees for Adam with a fixed step size. In **Section** 5, we present the details of the designed experiments and analyse the results. Finally, **Section** 6 concludes the article.

## 2 Notations and Definitions

In the following, we denote as $\mathbf{u}^T\mathbf{w}$ and $\|\mathbf{w}\|_2$ the scaler product and $L_2$-norm of the Hilbert space $\mathbb{R}^n$ respectively. For any differentiable and real-valued function $f : \mathbb{R}^n \mapsto \mathbb{R}$ and any point $\mathbf{w}_0 \in \mathbb{R}^n$, we will denote as $\nabla_\mathbf{w} f(\mathbf{w}_0) \in \mathbb{R}^n$ the gradient of $f$ at $\mathbf{w}_0$. The double derivative of $f$ at $\mathbf{w}_0$ is denoted by $\nabla_\mathbf{w}^2 f(\mathbf{w}_0) \in \mathbb{R}^{n \times n}$ which is a square matrix, also called the *Hessian matrix*. Next, we define the Lipschitz property, a standard assumption in optimisation literature, which we assume in all our proofs for non-convex objectives. Additionally, this section presents the pseudocode for the Adam optimizer.

**Definition 1. *K - Smoothness.*** *If $f : \mathbb{R}^d \mapsto \mathbb{R}$ is at least once differentiable, then we call it $K$ smooth for some constant $K > 0$, if for all $\mathbf{w}_1, \mathbf{w}_2 \in \mathbb{R}^d$, the following inequality holds:*

$$f(\mathbf{w}_2) \leq f(\mathbf{w}_1) + \nabla_{\boldsymbol{w}} f(\mathbf{w}_1)^T(\mathbf{w}_1 - \mathbf{w}_2) + \frac{K}{2}\|\mathbf{w}_1 - \mathbf{w}_2\|_2^2$$

If a function $f : \mathbb{R}^n \mapsto \mathbb{R}$ is $K$-smooth, then $K$ is essentially the Lipschitz constant of the gradient of $f$. Henceforth, we will refer to $K$ as the Lipschitz smoothness of the function, which is the same as the Lipschitz constant of the gradient of the function. Alternatively, we can define $K$-smoothness as:

**Definition 2. *K - Smoothness.*** *A function $f : \mathbb{R}^d \mapsto \mathbb{R}$ is called Lipschitz smooth if there exists a constant $K > 0$ such that:*

$$\forall \boldsymbol{w}_1, \boldsymbol{w}_2 \in \mathbb{R}^n, \|\nabla_{\boldsymbol{w}} f(\boldsymbol{w}_1) - \nabla_{\boldsymbol{w}} f(\boldsymbol{w}_2)\|_2 \leq K\|\boldsymbol{w}_1 - \boldsymbol{w}_2\|_2$$

---

[1]Better convergence refers to the gradient norm of the loss approaching zero more closely as iterations progress. If two algorithms, $\mathcal{A}_1$ and $\mathcal{A}_2$, drive the gradient norm of the loss to 0.1 and 0.001 respectively, $\mathcal{A}_2$ demonstrates superior convergence. To assess the convergence speed accurately, one must analyze the loss curve plot.

The smallest $K$ for which the previous inequality is true is called the Lipschitz constant of $\nabla f$ or Lipschitz smoothness of $f$ and will be denoted $K(f)$. Note, from here onwards $K$ and $K(f)$ can be used interchangeably, denoting the function's Lipschitz smoothness.

**Definition 3. *Square root of Penrose inverse.*** *If $\mathbf{v} \in \mathbb{R}^d$ and $\mathbf{V} = diag(\mathbf{v})$, then we define $\mathbf{V}^{\frac{-1}{2}} := \sum_{j \in support(\mathbf{v})} \frac{1}{\sqrt{v_j}} \mathbf{b}_j \mathbf{b}_j^T$, where $\{\mathbf{b}_j\}_{j=1}^d$ is the standard basis of $\mathbb{R}^d$.*

**Definition 4. *Sign function.*** *We define a function $sign : \mathbb{R}^d \mapsto \{-1, 1\}^d$ as follows:*

$$sign(\mathbf{z})_j = \begin{cases} 1 & if\ \mathbf{z}_j \geq 0 \\ -1 & else \end{cases}$$

## 3 Motivation & Contributions

In practical scenarios, such as learning latent variables from vast datasets with unknown distributions, the goal is to solve the optimisation problem:

$$\min_{\mathbf{w} \in \mathbb{R}^d} \mathcal{L}(\mathbf{w}) := \mathbb{E}_{\zeta \sim \mathcal{P}}[l(\zeta, \mathbf{w})] \tag{2}$$

Here $\mathcal{L} : \mathbb{R}^n \mapsto \mathbb{R}$ is a non-convex loss function, $\zeta$ is a random variable with an unknown distribution $\mathcal{P}$ and $l(\zeta, \mathbf{w})$ quantifies the performance of parameter vector $\mathbf{w} \in \mathbb{R}^n$ on the random variable $\zeta$. Common iterative optimisation algorithms like Adam and RMSProp are often employed to solve this problem. It's observed that using a more aggressive constant step size often leads to favourable outcomes (Jing et al., 2020; Li et al., 2021). However, determining the optimal learning rate requires an exhaustive search. Specifically, when optimizing for a parameter set $\mathbf{w} \in \mathbb{R}^d$, the goal is to find a parameter $\mathbf{w}^*$ aligning with a minimum in the loss landscape $\mathcal{L}(\mathbf{w})$. Gradient descent-based algorithms, including those with momentum, iteratively update parameters using $\mathbf{w}_{t+1} = \mathbf{w}_t - \alpha_t g(\nabla_{\mathbf{w}} \mathcal{L}(\mathbf{w}_t))$, where $\alpha_t$ is the step size at iteration $t$ and $g : \mathbb{R}^d \mapsto \mathbb{R}^d$ is a function incorporating the gradient and momentum terms. Convergence of the parameter sequence $\{\mathbf{w}_0, \mathbf{w}_1, \ldots, \mathbf{w}_{T-1}\}$ to $\mathbf{w}^*$ necessitates the sequence $\{(\mathbf{w}_{t+1} - \mathbf{w}_t)\}$ to converge to $\mathbf{0}$ as $t \to T$. This corresponds to the sequence $\{\alpha_t g(\nabla_{\mathbf{w}} \mathcal{L}(\mathbf{w}_t))\}$ converging to $\mathbf{0} \in \mathbb{R}^d$. The literature often opts for a non-increasing step size $\alpha_t$, such as $\alpha_t \propto \frac{1}{t^a}$, where $a > 0$, to theoretically prove convergence. Despite non-zero gradient norms, $\alpha_t$ can cause $\{\alpha_t g(\nabla_{\mathbf{w}} \mathcal{L}(\mathbf{w}_t))\}$ to approach $\mathbf{0}$, even if $\{\nabla_{\mathbf{w}} \mathcal{L}(\mathbf{w}_t)\}$ does not. Although accumulating past gradients helps mitigate the decay of the learning rate, there's a risk that learning rates with rapid decay may dominate. Our empirical analysis in **Section** 5 demonstrates that Adam, with non-increasing learning rates, does not aggressively drive the gradient norm of the loss towards zero. With a fixed step size $\alpha_t = \alpha > 0$, if $\mathbf{w}_t \to \mathbf{w}^*$, then $\|\nabla_{\mathbf{w}} \mathcal{L}(\mathbf{w}_t)\|_2 \to \mathbf{0}$. This ensures convergence to a saddle point of $\mathcal{L}(\mathbf{w})$ with a fixed step size, an assurance not available with a non-increasing and iteration-dependent step size.

Consequently, our research aims to identify the optimal constant step size, ensuring convergence in both deterministic and stochastic Adam iterations beyond a time threshold $t \geq T(\beta_1, \rho)$ where $T(\beta_1, \rho)$ is a natural number.

**A summary of our contributions**: The major contributions of this article are summarized as follows.

1. We derive an exact constant step size to guarantee convergence of deterministic as well as stochastic Adam. To the best of our knowledge, this study is the first to theoretically guarantee Adam's global convergence (gradient norm of loss function converges to 0) with an exact constant step size, which depends on the dynamics of the model and data.

2. Our study offers runtime bounds for deterministic and stochastic Adam to achieve approximate criticality with smooth non-convex functions.

3. We introduced a simple method to estimate the Lipschitz smoothness [2] of the loss function $w.r.t$ the network parameters. We offer a probabilistic guarantee for the convergence of our estimated Lipschitz constant to its true value.

---

[2]Our derived step size, crucial for ensuring convergence, is determined by the Lipschitz smoothness of the loss function.

4. We empirically show that even with past gradient accumulation, deterministic Adam can be affected by rapidly decaying learning rates. This indicates that these rapid decay rates play a dominant role in driving convergence.

5. We also demonstrate empirically that with our analysed step size, Adam quickly converges to a favourable saddle point with high test accuracy[3].

# 4 Convergence Guarantee For Adam With Fix Step Size

Previously, it has been shown that deterministic RMSProp and Adam can converge under certain conditions with adaptive step size (Sashank et al., 2018; De et al., 2018; Défossez et al., 2020). Here, we give the first result about convergence to criticality for deterministic and stochastic Adam with constant step size, albeit under a certain technical assumption about the loss function (and hence on the noisy first-order oracle). *The proofs of the theorems discussed in this section, along with additional experiments, are provided in the **Appendix**, which is included as part of the **Supplementary Material** for this paper.*

## 4.1 Novel Sufficient Conditions (SC) for Convergence of Adam

Below are the commonly employed assumptions for analyzing the convergence of stochastic algorithms for non-convex problems:

1. The minimum value of the problem $\mathcal{L}^* = argmin_{\mathbf{w} \in \mathbb{R}^d} \mathcal{L}(\mathbf{w})$ is lower bounded, *i.e* $\mathcal{L}^* > -\infty$.

2. The stochastic gradient $\mathbf{g}_t$ of the loss function is an unbiased estimate, *i.e* $\mathbb{E}[\mathbf{g}_t] = \nabla_{\mathbf{w}} \mathcal{L}(\mathbf{w}_t)$.

3. The second-order moment of stochastic gradient $\mathbf{g}_t$ is uniformly upper-bounded, *i.e* $\mathbb{E}[\|\mathbf{g}_t\|_2^2] \leq (\mathbb{E}[\|\mathbf{g}_t\|_2])^2 \leq \gamma^2$.

In addition, we suppose that that the parameters $\beta_1$ and learning rate $\alpha$ satisfy the following restrictions:

1. The parameter $\beta_1$ satisfies $\beta_1 < \frac{\epsilon}{\epsilon+\gamma}$ for some $\epsilon > 0$.

2. Our proposed step size $\alpha_t = \alpha \in \mathbb{R}^+$ remains constant throughout the analysis.

**Theorem 1.** ***Deterministic Adam converges with proper choice of constant step size.*** *Let the loss function $\mathcal{L} : \mathbb{R}^n \mapsto \mathbb{R}$ be $K-$smooth and let $\gamma < \infty$ be an upper bound on the norm of the gradient of $\mathcal{L}$. Also assume that $\mathcal{L}$ has a well-defined minimiser $\boldsymbol{w}^*$ such that $\boldsymbol{w}^* = argmin_{\boldsymbol{w} \in \mathbb{R}^d} \mathcal{L}(\boldsymbol{w})$. Then the following holds for Algorithm (1):*

*For any $\epsilon, \rho > 0$ if we let $\alpha = \sqrt{2(\mathcal{L}(\boldsymbol{w}_0) - \mathcal{L}(\boldsymbol{w}^*))/K\delta^2 T}$, then there exists a natural number $T(\beta_1, \rho)$ (depends on $\beta_1$ and $\rho$) such that $\|\nabla_{\boldsymbol{w}} \mathcal{L}(\boldsymbol{w}_t)\|_2 \leq \epsilon$ for some $t \geq T(\beta_1, \rho)$, where $\delta^2 = \frac{\gamma^2}{\rho^2}$.*

**Theorem 2.** ***Stochastic Adam converges with proper choice of constant step size.*** *Let the loss function $\mathcal{L} : \mathbb{R}^n \mapsto \mathbb{R}$ be $K-$smooth and be of the form $\mathcal{L} = \sum_{j=1}^m \mathcal{L}_j$ such that (a) each $\mathcal{L}_j$ is at-least once differentiable, (b) the gradients satisfy $sign(\mathcal{L}_r(\boldsymbol{w})) = sign(\mathcal{L}_s(\boldsymbol{w}))$ for all $r, s \in \{1, 2, \ldots, m\}$, (c) $\mathcal{L}$ has a well-defined minimiser $\boldsymbol{w}^*$ such that $\boldsymbol{w}^* = argmin_{\boldsymbol{w} \in \mathbb{R}^d} \mathcal{L}(\boldsymbol{w})$. Let the gradient oracle, upon invocation at $\boldsymbol{w}_t \in \mathbb{R}^d$, randomly selects $j_t$ from the set $\{1, 2, \ldots, m\}$ uniformly, and then provides $\nabla \mathcal{L}_{j_t}(x_t) = \boldsymbol{g}_t$ as the result.*

*Then, for any $\epsilon, \rho > 0$ if we let $\alpha = \sqrt{2(\mathcal{L}(\boldsymbol{w}_0) - \mathcal{L}(\boldsymbol{w}^*))/K\delta^2 T}$, then there exists a natural number $T(\beta_1, \rho)$ (depends on $\beta_1$ and $\rho$) such that $\mathbb{E}[\|\nabla_{\boldsymbol{w}} \mathcal{L}(\boldsymbol{w}_t)\|_2] \leq \epsilon$ for some $t \geq T(\beta_1, \rho)$, where $\delta^2 = \frac{\gamma^2}{\rho^2}$.*

**Remark 1**: With our analysis, we showed that both deterministic and stochastic Adam with constant step size converges with rate $\mathcal{O}\left(\frac{1}{T^{1/4}}\right)$. This convergence rate is similar to the convergence rate of SGD proposed in (Li et al., 2014). Our motivation behind these theorems was primarily to understand the conditions

---

[3]To convert the test accuracy on the y-axis to a percentage, multiply the y-axis values by 100.

under which Adam can converge, especially considering the negative results presented in (Sashank et al., 2018). However, we acknowledge that it is still an open problem to tighten the analysis of deterministic as well as stochastic Adam and obtain faster convergence rates than we have shown in the theorem. One thing to keep in mind is that our convergence rate is obtained for a fixed step size, and having a decaying step size provides additional opportunities for analysis to tighten the bound (Shi et al., 2020; Tian & Parikh, 2022; Chen et al., 2018; Luo et al., 2019; Défossez et al., 2020; Zou et al., 2019).

**Remark 2**: Based on the preceding two theorems, it becomes evident that the Adam achieves convergence when equipped with an appropriately chosen constant step size $\alpha = \sqrt{2(\mathcal{L}(\mathbf{w}_0) - \mathcal{L}(\mathbf{w}^*))/K\delta^2 T}$, **according to our analysis**. Subsequently, we analyse the given step size and conduct extensive experiments to ascertain that, with the designated step size, the gradient norm effectively and more aggressively tends towards zero compared to other learning rate scheduling techniques (listed in **Table** 2 in **Appendix**). Hence, our empirical investigations validate that the chosen step size ensures convergence in practice. We compare our step size with several state-of-the-art schedulers and an array of constant step sizes in **Section** 5.

## 4.2 Analysis of Constant Step Size

The optimal learning rate, as per **Theorem** 1 and **Theorem** 2, depends on factors like the Lipschitz smoothness of the loss[4] ($K$), initial and final loss values, the total number of iterations/epochs ($T$), and an additional term denoted as $\delta^2$. For simplicity, we omitted the $\delta^2$ term as it depends on an oracle of gradients and set the final loss term to zero. Thus, our approximate learning rate is now expressed as $\alpha_{ours} \approx \sqrt{\frac{2\mathcal{L}(\mathbf{w}_0)}{\hat{K}T}}$, making it practical while still capturing the core concept of **Theorem** 1 and **Theorem** 2 . Here $\hat{K}$ is the approximated smoothness of the loss, defined in the next section.

## 4.3 Approximating The Lipschitz Smoothness of Loss

Determining the appropriate Lipschitz smoothness for a neural network is a key area of research in deep learning, with various methods proposed for estimation, as demonstrated in recent works (Virmaux & Scaman, 2018; Fazlyab et al., 2019a; Latorre et al., 2020; Fazlyab et al., 2019b; Gouk et al., 2021). Estimating the learning rate prior to training requires careful estimation of the Lipschitz smoothness of the loss function. For locally Lipschitz smooth functions (i.e. functions whose restriction to some neighbourhood around any point is Lipschitz), the Lipschitz smoothness may be computed using its Hessian matrix.

**Theorem 3.** *(Rademacher (Federer, 2014) [22, Theorem 3.1.6]) If $f : \mathbb{R}^d \mapsto \mathbb{R}$ is a locally Lipschitz smooth function, then $f$ is differentiable almost everywhere. Moreover, if $f$ is at least twice differentiable and $\nabla_{\mathbf{w}} f$ is Lipschitz continuous, then*

$$K(f) = sup_{\mathbf{w} \in \mathbb{R}^d} \|\nabla_{\mathbf{w}}^2 f\|_2 \tag{3}$$

*where $\|\mathbf{A}\|_2$ is the spectral norm of the matrix $\mathbf{A} \in \mathbb{R}^{n \times n}$.*

For large networks, directly optimizing according to **Theorem** 3 can be time-consuming. Instead, we propose a novel technique to approximate the Lipschitz smoothness. It's important to note that this work does not aim to estimate the best Lipschitz smothness for the loss or network; rather, it focuses on the convergence analysis of Adam. We introduce Algorithm 2 to approximate the Lipschitz smoothness of the loss[5], supported by mathematical proof that our estimate *converges in distribution* to the original Lipschitz smoothness. We shall now prove that our estimated Lipschitz smoothness ($\hat{K}$) converges to the real Lipschitz smoothness ($K$) in distribution.

**Theorem 4.** *Given $\|.\|_2$ and $\nabla_{\mathbf{w}}^2 \mathcal{L}$ are continuous functions and $\|\nabla_{\mathbf{w}}^2 \mathcal{L}\|_2$ is bounded from above. If $\mathcal{L}$ is locally Lipschitz smooth then, $K(\mathcal{L}) = sup_{\mathbf{w} \in \mathbb{R}^d} \|\nabla_{\mathbf{w}}^2 \mathcal{L}\|_2$. Let $\hat{K}$ be a random variable defined as :*

$$\hat{K} = \max_{1 \leq j \leq N} \|\nabla_{\mathbf{w}}^2 \mathcal{L}(\mathbf{w}_j)\|_2 \tag{4}$$

---

[4]The Lipschitz smoothness of the loss is with respect to network parameters.
[5]**Algorithm** 2 is given for the full-batch scenario. The algorithm for the mini-batch case is deferred to the **Appendix** due to space constraints.

where $\boldsymbol{w}_j$, $j \in \{1, 2, \ldots, N\}$ *are i.i.d samples drawn from a full domain distribution* $\mathcal{W}$*. Then, as* $N \to \infty$*, the random variable* $\hat{K}$ *converges in distribution to* $K(\mathcal{L})$*.*
*Mathematically:*

$$\lim_{N \to \infty} P(\hat{K} \leq k) = \begin{cases} 0 & if \ k < K(\mathcal{L}) \\ 1 & if \ k \geq K(\mathcal{L}) \end{cases} \tag{5}$$

---

**Algorithm 2:** Estimating Lipschitz Smoothness of Loss Function

---

**Input:** Dataset: $\mathcal{X} \sim \mathcal{P}$, Loss function: $\mathcal{L} : \mathbb{R}^d \to \mathbb{R}$, A network parameterized by $\mathbf{w} \in \mathbb{R}^d \sim \mathcal{W}$, No. of iterations: $N$

**Output:** $\hat{K}$

**1 Initialization:** $\hat{K} = 0$, Iteration Number $n = 0$

**2 for** $n$ *from 1 to* $N$*:* **do**

**3**   Randomly sample the network weights: $\mathbf{w}_n \sim \mathcal{W}$

**4**   Compute loss for the current pass: $\mathcal{L}(\mathbf{w}_n) = \mathbb{E}_{\zeta \sim \mathcal{P}}[l(\zeta, \mathbf{w}_n)]$

**5**   Compute the spectral norm of hessian at the current instant using power method (Müntz et al., 1913): $\|\nabla_{\mathbf{w}}^2 \mathcal{L}(\mathbf{w}_n)\|_2$

**6**   Estimate the Lipschitz smoothness $(\hat{K})$: $\hat{K} = max(\|\nabla_{\mathbf{w}}^2 \mathcal{L}(\mathbf{w}_n)\|_2, \hat{K})$

**7 End**

---

From **Theorem** 4, $\hat{K} \xrightarrow{d} K(\mathcal{L})$[6]. optimisation of the loss gradient can be achieved through iterative techniques like steepest descent (Boyd & Vandenberghe, 2004), backtracking (Boyd & Vandenberghe, 2004), and Wolfe's and Goldstein's conditions (Wolfe, 1969), aiming to identify the direction (weights $\mathbf{w}$) maximizing the gradient norm. However, non-convex loss surfaces in deep learning may lead to convergence to sub-optimal local maxima. Also, losses with unbounded gradients can lead to excessively high Lipschitz constants. Our approach addresses this by leveraging the *convergence in distribution* phenomenon from probability theory. We randomly select directions from a distribution $\mathcal{W}$ and take the maximum gradient norm across all evaluated samples, theoretically converging to the true Lipschitz constant in distribution. Note, to respect **Theorem** 3, the distribution of the weights $\mathcal{W}$ needs to be full domain.

## 5 Experimental Setup

We assess Adam's performance with our selected step size through experiments on fully connected networks with ReLU activations. Additionally, we expand our analysis to include classification tasks on MNIST and CIFAR-10 datasets using LeNet (LeCun et al., 1998) and VGG-9 (Simonyan & Zisserman, 2014), respectively. To maintain strict experimental control, we ensure that all network layers have identical widths denoted as $h$. For experiments, in **Section** 5.2 and 5.3, we used Kaiming uniform distribution (He et al., 2015) for initialization. Default PyTorch parameters are chosen [Link]. No regularization is applied. Before delving into further details about the experiments, we will first describe the model architectures utilized for experiments in the subsequent sections.

### 5.1 Outline of the Experiments Conducted

We conducted classification experiments on the MNIST dataset for various network sizes, encompassing different values of network depth and $h$. CIFAR-10 experiments are carried out only on VGG-9 architecture. All experiments are implemented using PyTorch. We compare the performance of Adam using **our proposed step size** with:

1. A list of commonly used schedulers and several non-increasing step sizes that were utilized to demonstrate theoretical convergence for Adam in past literature. The schedulers include **(i)** Linear LR,

---

[6]**Theorem** 4 holds for mini-batch case too. Refer to **Appendix** for more insights.

**(ii)** Step LR, **(iii)** Square root Decay, **(iv)** Inverse Time Decay, **(v)** Exponential Decay, **(v)** One Cycle LR, and **(vi)** Cosine Decay (Smith & Topin, 2019). The full list with hyperparameters are given in **Table** 2, **Appendix** C.2.

2. An array of constant step sizes $\{10^{-1}, 10^{-2}, 10^{-3}, 10^{-4}, 10^{-5}\}$ which are widely used in literature. Additionally, we conducted a fine search around our proposed step size $\alpha_{\text{ours}}$ by varying it by a factor of 2. The final comparison is made using the list $\{10^{-1}, 10^{-2}, 10^{-3}, 10^{-4}, 10^{-5}, \frac{\alpha_{\text{ours}}}{2}, 2\alpha_{\text{ours}}\}$.

Each experiment was executed for 100 epochs. For various schedulers, we fine-tuned their hyperparameters to identify configurations where they exhibited the best performance and retained those values for the final comparison with our step size. These choices are made to provide a fair comparison with our constant step size approach, as overly rapid decay rates may not effectively minimise the gradient norm of the loss function. Below is the list of experiments we have conducted:

1. **Full-batch experiments (Section 5.2)**: To efficiently compare learning rate schedules, we created smaller versions of the MNIST and CIFAR-10 datasets: mini-MNIST and mini-CIFAR-10, respectively. Mini-MNIST comprises 5500 training images and 1000 test images, selected from the original MNIST dataset. Mini-CIFAR-10 includes 500 training images and 100 test images per class, representing about 10% of the full CIFAR-10 dataset. Despite their reduced size, both mini datasets adequately represent their respective originals for experimentation.

2. **Mini-batch experiments (Section 5.3)**: To see if our findings from the full-batch experiments apply to mini-batch scenarios according to **Theorem** 2, we conducted similar experiments using a mini-batch setup with a fixed batch size of 5,000 for CIFAR10 and MNIST both. We utilized the entire training sets of CIFAR10 and MNIST for these experiments. We tested our models on CIFAR10 using the VGG-9 architecture and on MNIST using the LeNet architecture. We also justified why such a large batch size is needed for empirical convergence.

3. **Additional experiments on CNN architectures (Section 5.4)**: To examine the potential generalizability of our theoretical findings across architectures, we trained an image classifier on CIFAR-10 and MNIST datasets employing VGG-like networks and LeNet architectures, respectively.

4. **Effect of varying the training horizon** ($T$) **(Section 5.5)**: Our proposed learning rate is dependent on the training horizon ($T$). We provide empirical evidence showing that tuning the training horizon preserves the optimization superiority of our fixed learning rate. Since the training horizon is a critical hyperparameter in practical applications (Dahl et al., 2023), understanding its impact is essential for the success of any new optimization methods.

5. **Effect of varying $\rho$ (Section 5.6)**: The parameter $\rho$ is a small positive value, serving as a hyperparameter in the Adam optimization algorithm. It is introduced to prevent the gradient accumulation term from blowing up, ensuring numerical stability during training. In our experiments, we fine-tune this parameter to evaluate its impact on the performance of our proposed learning rate.

6. **Effect of initialization on our learning rate**: As our learning rate $\alpha_{ours} = \sqrt{\frac{2\mathcal{L}(\mathbf{w}_0)}{\hat{K}T}}$ depends on the initial loss value and the estimated Lipschitz constant, we conducted experiments to demonstrate the independence of our learning rate with respect to different network initialization techniques. *Please refer to* **Appendix** C.5 *for the results.*

## 5.2 Comparing Performances in Full-Batch Setting

In **Figure** 1 and 2, we illustrate the variations in gradient norm, training loss, and test loss across iterations for different schedulers. These experiments were conducted on single-layer, three-layer, and five-layer classifiers with 1000 nodes in each hidden layer, trained on mini-MNIST. Additional comparisons for various neural network architectures with different widths and depths are provided in the **Appendix**, where similar qualitative trends can be observed. **(1) Comparison against various learning rate schedulers**:

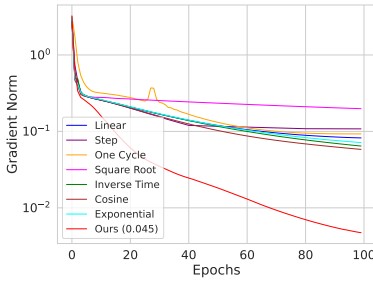 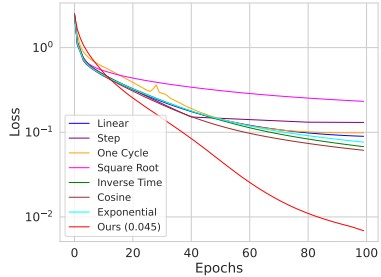 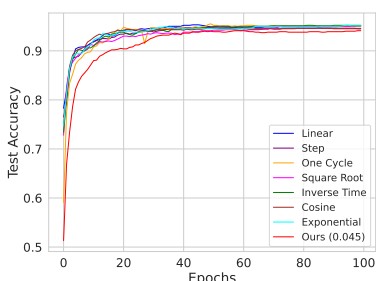

Figure 1: *Comparison with schedulers*: **Full-batch** experiments on a 3-layer network with $10^3$ nodes/layer, trained on MNIST. *Left*: Grad. norm v/s epochs, *Middle*: Training loss v/s epochs, *Right*: Test acc. v/s epochs.

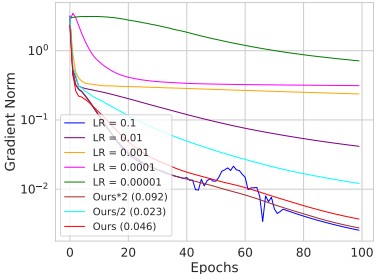 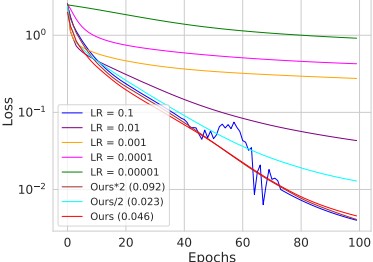 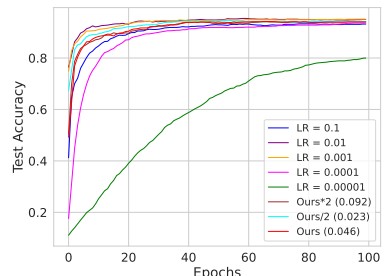

Figure 2: *Comparision with const. step sizes*: **Full-batch** experiments on a 3-layer network with $10^3$ nodes/layer, trained on MNIST. *Left*: Grad. norm v/s epochs, *Middle*: Training loss v/s epochs, *Right*: Test acc. v/s epochs.

We compared our step size against several well-known schedulers listed in **Table** 2, **Appendix** C.2. Except for cosine decay and one-cycle learning rate, all other schedulers are non-increasing. Our goal was to evaluate their efficacy in driving the gradient norm of the loss function towards zero. **(2) Comparison against various fixed learning rates**: We tested our learning rate against several other fixed learning rates, $\{10^{-1}, 10^{-2}, 10^{-3}, 10^{-4}, 10^{-5}, 2 \times \alpha_{ours}, \frac{\alpha_{ours}}{2}\}$. We wanted to demonstrate that instead of searching extensively for the best learning rate to train a model, one could just use our learning rate and achieve decent results.

Looking at **Figure** 1, it is clear that our suggested learning rate helps decrease the gradient norm of loss more aggressively towards zero as compared to other schedulers, supporting our **Motivation**. Even with the accumulation of a few past gradients, deterministic Adam proves ineffective in mitigating the rapid decay of the learning rate. This is evident from **Figure** 1, where one can observe a notable difference in gradient norms after 100 epochs between the best-performing scheduler (Exponential in this case) and our learning rate is in order of $10^{-2}$.

From **Figure** 2, we observe that our proposed step size effectively reduces the gradient norm of the loss function. Although the constant learning rate of $10^{-1}$ achieves a slightly lower gradient norm than our method (the difference is minimal and nearly negligible), it shows noticeable jerks during training. This pattern is also evident in other figures, such as **Figure** 4, **Figure** 8, and **Figure** 11. In **Figure** 8 and **Figure** 11, the jerks are more pronounced compared to **Figure** 2. This is because, in CNNs, which are more complex models than small fully connected networks, finding local minima is more challenging with such a large learning rate. The learning rate of $10^{-1}$ is relatively high for training neural networks, which can lead to overshooting of local minima or maxima. In contrast, our learning rate decreases both the loss and its gradient norm smoothly during training.

Additionally, these figures demonstrate that our learning rate not only reaches any stationary point but tends to approach a local minimum or its neighbourhood where the model achieves good test accuracy[7]. The plot of loss versus epoch indicates that our proposed learning rate leads to faster convergence in practice as

---

[7]The test accuracy on the y-axis is **NOT** in percentage. To convert it to percentage, multiply the y-axis values by 100.

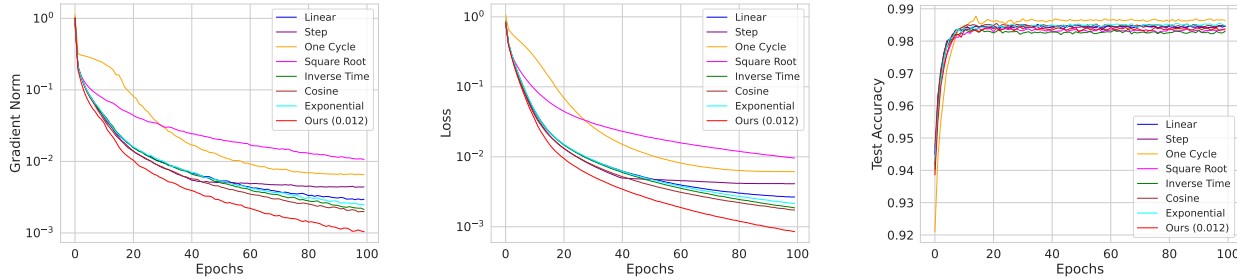

Figure 3: *Comparison with schedulers*: **Mini-batch** experiments on a 3-layer network with $10^3$ nodes/layer, trained on MNIST. *Left*: Grad. norm v/s epochs, *Middle*: Training loss v/s epochs, *Right*: Test acc. v/s epochs.

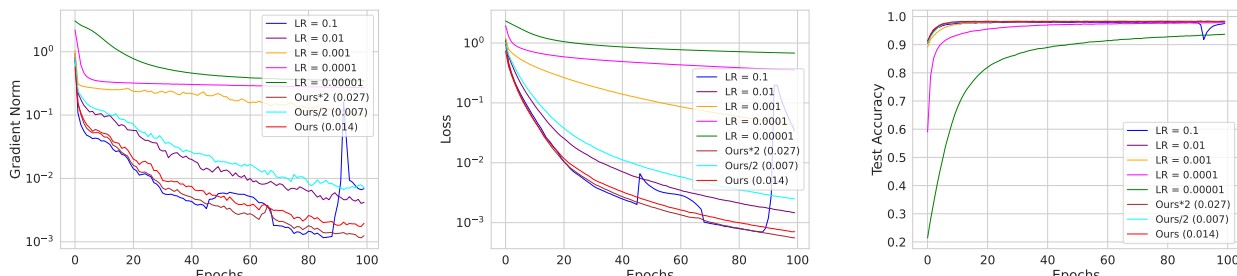

Figure 4: *Comparison with const. step sizes*: **Mini-batch** experiments on a 1-layer network with $10^3$ nodes/layer, trained on MNIST. *Left*: Grad. norm v/s epochs, *Middle*: Training loss v/s epochs, *Right*: Test acc. v/s epochs.

compared to other schedulers. However, it's worth noting that achieving faster convergence in general is still a challenge. Nevertheless, using this learning rate results in quicker convergence than other schedulers and a range of fixed learning rates.

## 5.3 Comparing Performances in Mini-Batch Setting

In this section, we replicate the same set of experiments in mini-batch setting. We choose 5,000 as batch size for both the CIFAR10 and MNIST datasets. From **Figure** 3 and **Figure** 4, we observed that even in the mini-batch setting, our learning rate effectively reduces the gradient norm of loss more aggressively compared to other schedulers and constant step sizes. This trend aligns with the behaviour observed in the full-batch setting across all plots, including gradient norm, training loss, and test accuracy.

**Why such a large batch size is chosen?** Despite the non-convergence issue (Sashank et al., 2018), it does not rule out convergence if the minibatch size increases over time, thereby reducing the variance of stochastic gradients. Increasing minibatch size has been shown to aid convergence in some optimisation algorithms not based on *exponential moving average* (EMA) methods (Bernstein et al., 2018; Hazan et al., 2015). In all our minibatch experiments (including those in the **Appendix**), we choose a large batch size because our theoretical analysis for the minibatch setting, as described in **Theorem** 2, assumes that the loss is $K - Lipschitz$ and is represented by $\mathcal{L} = \sum_{j=1}^{m} \mathcal{L}_j$. Our proposed learning rate relies on the Lipschitz constant of the loss function. To accurately estimate this constant, it is crucial to estimate the stochastic gradients with low variance, as the Lipschitz constant is essentially the supremum of the true gradient norm over the network weights (**Theorem** 3). This approximation of gradients affects the estimation of the Lipschitz constant. Therefore, to make our step size effective in the minibatch setting, it is crucial to estimate the gradients with low variance, which in turn provides the correct estimation of the Lipschitz constant. From **Figure** 5, our empirical analysis suggests that increasing batch size will lead to faster convergence, as increasing batch size decreases variance. We now show results with our learning rate with increasing minibatch size in **Figure** 5.

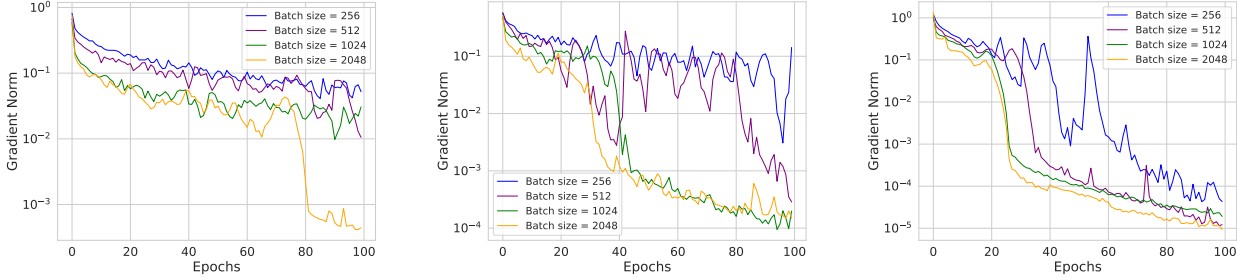

Figure 5: Increasing batch size experiments on *Left*: A fully connected network with 5 layers and $3 \times 10^3$ nodes/layer, trained on MNIST, *Middle*: LeNet trained on MNIST and *Right*: VGG-9 trained on CIFAR-10.

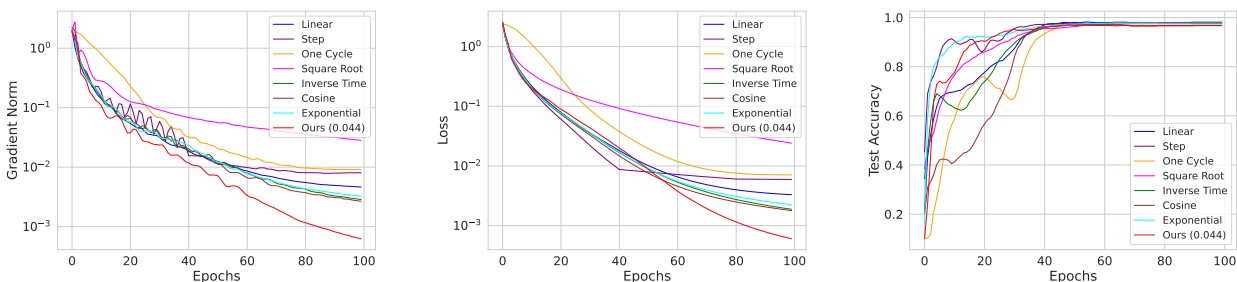

Figure 6: *Comparison with schedulers:* **Full-batch** experiments on LeNet architecture with MNIST data. *Left*: Grad. norm v/s epochs, *Middle*: Training loss v/s epochs, *Right*: Test acc. v/s epochs.

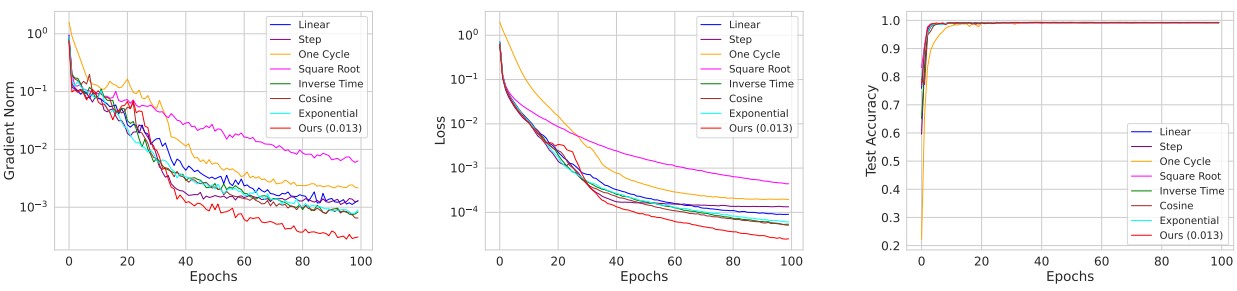

Figure 7: *Comparison with schedulers:* **Mini-batch** experiments on LeNet architecture with MNIST data. *Left*: Grad. norm v/s epochs, *Middle*: Training loss v/s epochs, *Right*: Test acc. v/s epochs.

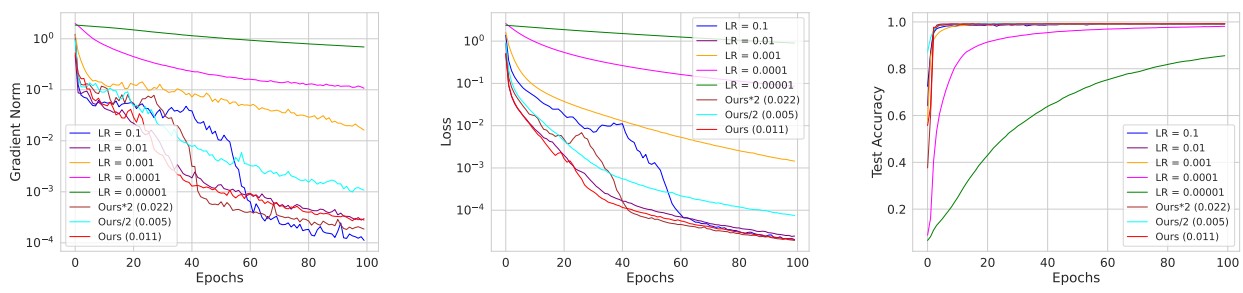

Figure 8: *Comparison with constant step sizes:* **Mini-batch** experiments on LeNet architecture with MNIST data. *Left*: Grad. norm v/s epochs, *Middle*: Training loss v/s epochs, *Right*: Test acc. v/s epochs.

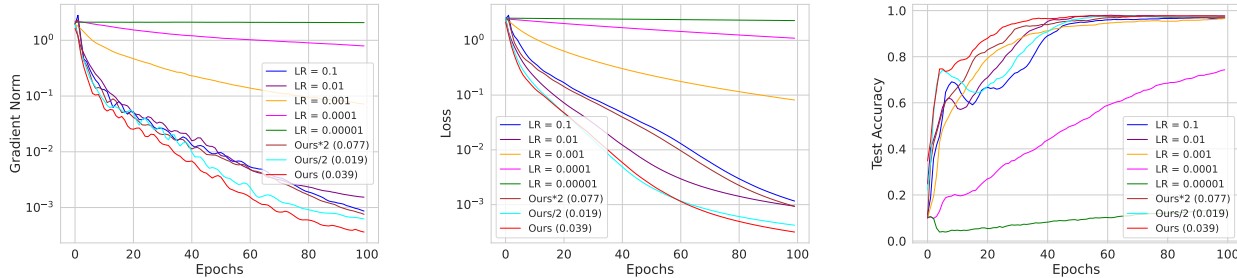

Figure 9: *Comparison with constant step sizes:* **Full-batch** experiments on LeNet architecture with MNIST data. *Left*: Grad. norm v/s epochs, *Middle*: Training loss v/s epochs, *Right*: Test acc. v/s epochs.

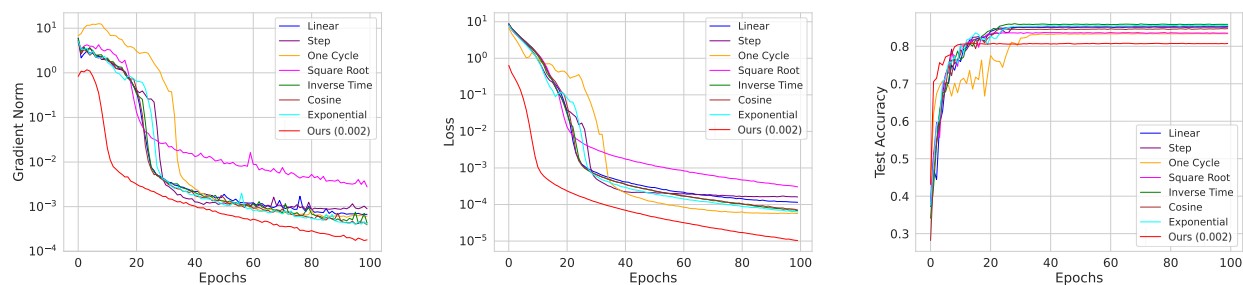

Figure 10: *Comparison with schedulers:* **Mini-batch** experiments on VGG-9 architecture with CIFAR-10 data. *Left*: Grad. norm v/s epochs, *Middle*: Training loss v/s epochs, *Right*: Test acc. v/s epochs.

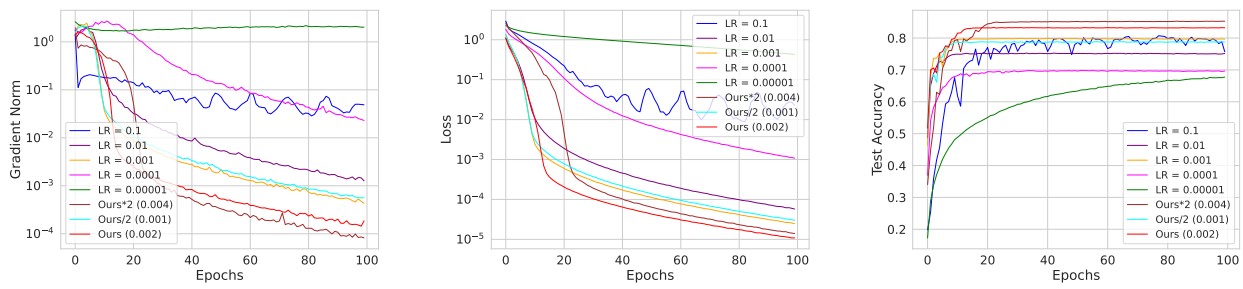

Figure 11: *Comparison with constant step sizes:* **Mini-batch** experiments on VGG-9 architecture with CIFAR-10 data. *Left*: Grad. norm v/s epochs, *Middle*: Training loss v/s epochs, *Right*: Test acc. v/s epochs.

### 5.4 LeNet on MNIST and VGG-9 on CIFAR-10

To test whether these results might qualitatively hold for convolutional neural networks (CNN) models, we trained an image classifier on CIFAR-10 and MNIST using VGG-9 and LeNet, respectively. We use minibatches of size 5,000 for the LeNet and 2,500 for VGG-9. As illustrated in **Figures** 6, 7, 8, 9, 10, and 11, our proposed learning rate scheduler performs well with CNNs, outperforming state-of-the-art schedulers and various constant step sizes. We can clearly see a gap of at least $10^{-1}$ order between our learning rate and the best-performing scheduler's gradient norm curve for LeNet experiments. This training was conducted for 100 epochs. Therefore, our empirical results demonstrate that our proposed learning rate can be effectively applied to CNNs and mini-batch scenarios, leading to better convergence.

### 5.5 Effect of Varying the Training Horizon ($T$)

Our proposed learning rate depends on the training horizon ($T$), which plays a crucial role in determining its overall effectiveness. In our experiments, we provide empirical evidence demonstrating that tuning the training horizon preserves the convergence superiority of our fixed learning rate. Specifically, adjusting the training horizon has a profound effect on how well the learning rate performs across different stages of the optimization process, ensuring stable and consistent learning throughout. This observation is particularly important, as the training horizon is widely recognized as a critical hyperparameter in many practical machine learning applications (Dahl et al., 2023). A thorough understanding of the impact of the training horizon is essential for successfully implementing new optimization methods, especially when aiming to generalize across a wide variety of tasks and datasets. It affects how the model converges and the quality of the final solution.

To further emphasize the robustness of our fixed learning rate, we have presented **Figures** 12, 13, 14 and 15 illustrating the relationship between the gradient norm of the loss over different training epochs—specifically at $20, 50$, and $150$ epochs. These plots clearly show that our learning rate performs consistently across all epochs, exhibiting stable optimization behavior regardless of the chosen training horizon. This consistency under varying training lengths highlights the adaptability of our approach, making it a reliable choice for different training scenarios and durations.

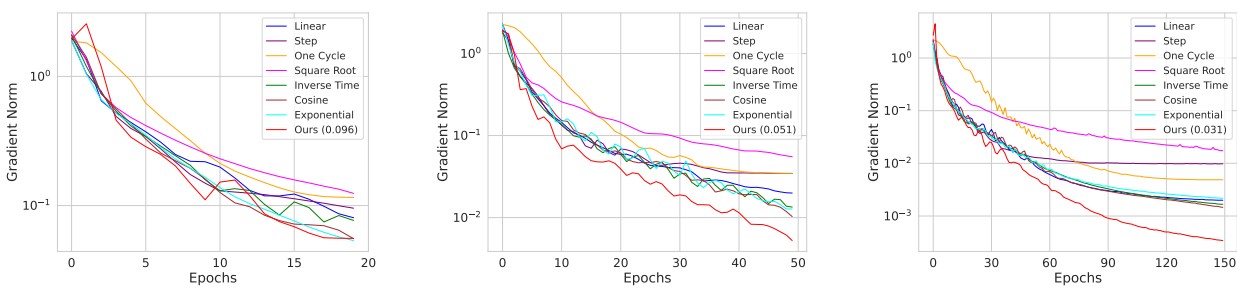

Figure 12: *Comparison with schedulers*: **Full-batch** experiments on LeNet and MNIST with varying training horizon ($T$). *Left*: $T = 20$, *Middle*: $T = 50$, *Right*: $T = 150$.

### 5.6 Effect of Varying $\rho$

Through this tuning process, we aim to better understand how $\rho$ influences the optimization dynamics and how it interacts with our learning rate strategy to ensure stable and efficient training. From **Figure** 16, we observe a slight overshoot in the gradient norm and loss for higher values of $\rho$. As $\rho$ decreases, the training process tends to become smoother. Since $\rho$ is used solely to prevent the gradient accumulation term $\mathbf{V}^{-1/2}$ from exploding due to zero values, setting $\rho$ too high may affect the optimization performance of Adam. For consistency, in all of our experiments, we have used the default value of $\rho = 10^{-8}$, as provided in the PyTorch package.

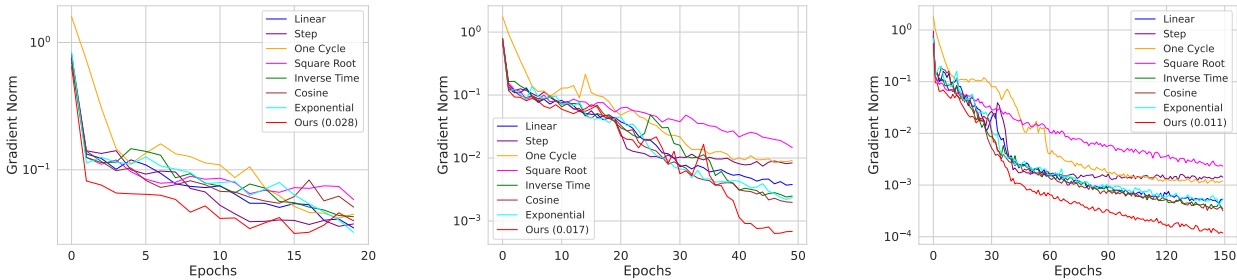

Figure 13: *Comparison with schedulers*: **Mini-batch** experiments on LeNet and MNIST with varying training horizon ($T$). *Left*: $T = 20$, *Middle*: $T = 50$, *Right*: $T = 150$.

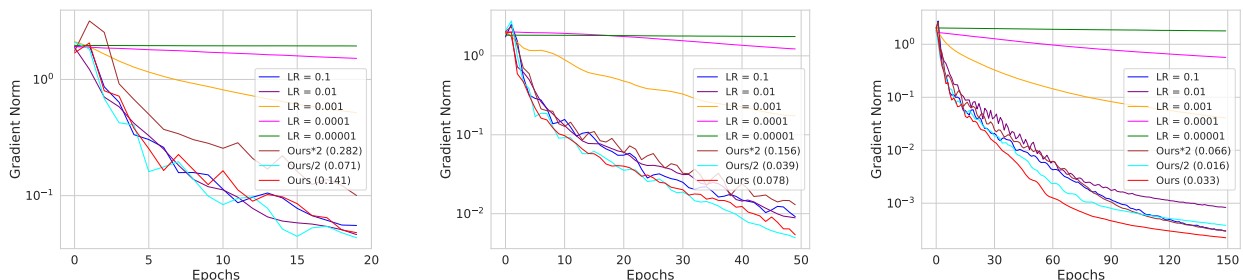

Figure 14: *Comparison with constant step sizes*: **Full-batch** experiments on LeNet and MNIST with varying training horizon ($T$). *Left*: $T = 20$, *Middle*: $T = 50$, *Right*: $T = 150$.

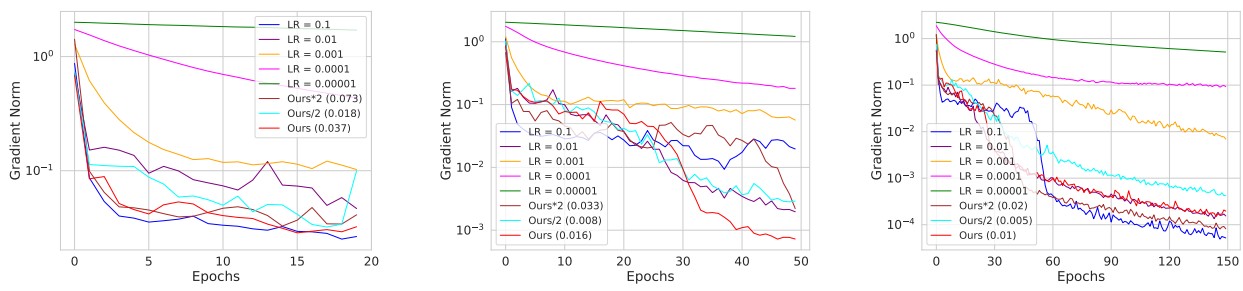

Figure 15: *Comparison with constant step sizes*: **Mini-batch** experiments on LeNet and MNIST with varying training horizon ($T$). *Left*: $T = 20$, *Middle*: $T = 50$, *Right*: $T = 150$.

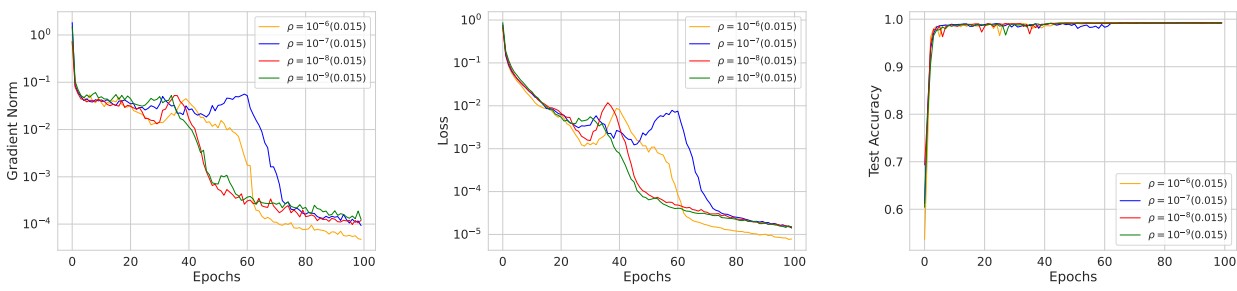

Figure 16: Effect of Tuning $\rho$ in LeNet Architecture on MNIST Dataset and observing it's effect on our proposed learning rate.

## 6  Conclusion and Limitations

This article presents the first known theoretical guarantees for the convergence of Adam with an exact constant step size in non-convex settings. It provides insights into why using non-decreasing step sizes may yield sub-optimal results and advocates for sticking to a fixed step size for convergence in Adam. **(i)** We analysed Adam's convergence properties and established a simple condition on the step size that depends on the characteristics of the model and data (in the form of the loss's Lipschitz smoothness and initial loss value). This condition ensures convergence in both deterministic and stochastic non-convex settings. We tag it as a sufficient condition **SC** in our text. **(ii)** We proposed a novel method for efficiently approximating the Lipschitz constant of the loss function with respect to the parameters, which is crucial for our proposed learning rate. **(iii)** Our empirical findings suggest that even with the accumulation of the past few gradients, the key driver for convergence in Adam is the non-increasing nature of step sizes. **(iv)** Finally, our theoretical claims are validated through extensive experiments on training deep neural networks on various datasets. Our experiments validate the effectiveness of the proposed step size. It drives the gradient norm towards zero more aggressively than the commonly used schedulers and a range of arbitrarily chosen constant step sizes. Our derived step size is easy to use and estimate and can be used to train a wide range of tasks.

**Limitations**: **(i)** It is still open to tightening the convergence rate of Adam. **(ii)** In **Section** 5.3, we empirically demonstrate that increasing the batch size leads to improved convergence. In our proof for the convergence of stochastic Adam, we chose to exclude the batch size factor for simplicity. However, we aim to theoretically demonstrate in the future that, according to our analysis, increasing batch size results in faster convergence.

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
