# OpenReview forum: "A Theoretical and Empirical Study on the Convergence of Adam with an “Exact" Constant Step Size in Non-Convex Settings"
_TMLR — Rejected by TMLR_

### Review · Reviewer_JZNx · 2024-08-25

**Summary Of Contributions:**

The authors study the convergence properties of Adam in full batch and minibatch settings. In each of these settings they prove that, with the appropriate lipschitz constant and bounded gradients, Adam with a particular constant stepsize will eventually drive gradients to be small. They then train networks on subsets of CIFAR and MNIST with Adam using the theoretically predicted stepsize and show that in this setting, constant stepsize training optimizes better, but generalizes worse, than a particular sweep over base learning rates, even with schedules.

**Audience:**

No

**Claims And Evidence:**

No

**Requested Changes:**

Major:

I would like to see a more precise definition for the theorem so I can try to understand it better.

All experimental sweeps should include values larger than $10^{-2}$, and have grid spacing of factor of 2 or 3 (at least near the optimal learning rates).

Additional experiments with varying T should be conducted and put in the main text.

Minor:

In the abstract, remove the (i), (ii), etc, they don't add much.

In Equation 1, what is v?

If vectors are boldface, matrices should be as well.

Some of the references are poorly formatted; likely a bibliography settings issue.

**Strengths And Weaknesses:**

The paper tackles an interesting question about the relationship between Adam, stepsize, and curvature. It seems to build appropriately on previous works to do so.

However, I do not understand the main theorems. In particular, the useage of $T$ and $t$ is quite confusing to me. $T$ shows up as a constant in the stepsize, and then as a function of parameters; $t$ is used to index over the set $\\{1,\cdots, T\\}$ in the minimization operation, but then it is also claimed that the minimized criterion holds for $t \geq T$. I am hoping the authors or other reviewers can clarify this issue for me; it is quite possible that I missed something simple.

More broadly, I'm not sure how relevant this notion of convergence is in the non-convex setting. The local geometry of loss landscapes in deep learning can change quite rapidly. One of the effects of this is that certain optimizers can get to regions with small gradients, but they are still far from optimal (and indeed have poor performance). This can be seen even when comparing SGD and Adam in NLP tasks in the large/full batch setting [4]. This is reflected in the loss trajectories which slow down very rapidly for SGD; for full batch training, $d\mathcal{L}/dt$ is proportional to the gradient norm squared. Therefore SGD appears to ``converge'' better and faster according to the definition in this paper, when in reality it is perhaps just optimizing poorly.

There are also issues with the form of the stepsize. For one, the maximum value of the Hessian (and therefore the local lipschitz constant) can vary wildly for neural networks over initialization space, particularly in the limit of large dimensions or with ReLU activations. Indeed, even for a training trajectory we can observe massive increases in the maximum Hessian eigenvalue [2]. This increasing phenomenon can be even worse in Adam [3]. This could lead to very small values of $K$ and therefore small stepsize; this suggests that the form of the stepsize is very sensitive to estimation of K. Indeed, there are many situations where conservative estimation of $K$ can lead to more practical stepsize recommendations than more accurate ones.

The stepsize also depends on $\rho/\sqrt{T}$. In many settings there are a wide range (at least a factor of 10) of $\rho$ which train well, with similar stepsizes. The total training time $T$ may also vary widely during an experimental run. This would suggest that the proposed stepsize is also sensitive to the exact value of $\rho$ and $T$, and this can also cause deviation away from good stepsizes found by tuning.

I have major concerns with the experiments. The first is that the theoretically predicted learning rate is larger than $10^{-2}$ (I think this is the case; I could not find the actual value listed anywhere). However, the sweep has **maximum** learning rate $10^{-2}$. This means that the other settings all have learning rate less than the theoretical stepsize at all steps. This makes the experiments at best, inconclusive.

There is also a question of setting; the mini-CIFAR10 and the mini-MNIST are both very small datasets, and it is unclear what general lessons we can draw from this setup. Additionally, in both of these settings with any reasonable network one quickly reaches interpolation, and generalization matters far more than optimization/training performance - where, to my eye, the other methods have a distinct advantage. Additionally, SGD is known to generalize better than Adam on CIFAR10 scale datasets using architectures like ResNet.

Another issue with the experiments is that the theoretical learning rate depends on the training horizon $T$, but the paper didn't present empirical evidence that tuning the horizon maintains even the optimization superiority of the theoretical fixed learning rate. Training horizon is an important hyperparameter to tune in practical settings [5], so understanding this is important to any new methods.

Overall I also don't understand the point of the comparison of learning rate schedules to the theoretically chosen fixed learning rate. In all  major applications, Adam is paired with either a decaying learning rate schedule or an averaging scheme; constant learning rate schedules just don't work well on their own.

For these reasons I have concerns about both the validity of the work, and its overall interest to TMLR readership.

[1] https://arxiv.org/abs/2206.13424

[2] https://openreview.net/forum?id=jh-rTtvkGeM

[3] https://arxiv.org/abs/2207.14484

[4] https://arxiv.org/abs/2304.13960

[5] https://arxiv.org/abs/2306.07179

---

> ### Author Response · Authors · 2024-09-17
> **Response to reviewer jZNx**
>
> Thank you for your valuable comments. We have addressed weaknesses and revised the manuscript, highlighting the modifications in red according to the required changes. Please review the updated manuscript for detailed changes
>
> **1. However, I do not understand the main theorems. In particular, the usage of T and t  is quite confusing to me. T  shows up as a constant in the stepsize, and then as a function of parameters; t  is used to index over the set {1,⋯,T}  in the minimization operation, but then it is also claimed that the minimized criterion holds for t≥T. I am hoping the authors or other reviewers can clarify this issue for me; it is quite possible that I missed something simple.**
>
> _Response:_ When we discuss ensuring convergence in **Theorems 1 and 2**, we refer specifically to the convergence where the 2-norm of the gradient of the loss function approaches zero, which is a common measure of convergence, although there are other definitions. In this work, we derived a constant step size that guarantees convergence of Adam by driving the gradient norm of the loss to zero after a certain number of iterations  $T(\beta_1, \rho)$.
>
> Our research focuses on identifying the optimal constant step size that ensures convergence in both deterministic and stochastic Adam iterations beyond a time threshold $t \geq T(\beta_1, \rho)$, where $T(\beta_1, \rho)$ is a natural number. The index over the set mentioned is a part of our proof and should not appear in the theorem's main statement. It was a typo in the original manuscript that we have corrected in our revised manuscript.
>
> I hope this clarifies your doubts, and we are happy to address any further questions you may have.

---

> > ### Comment · Reviewer_JZNx · 2024-09-17
> > **Thanks for clarification on proof notation**
> >
> > Thank you for improving the notation in the proofs; this clarifies the setting for me.

---

> ### Author Response · Authors · 2024-09-17
> **Response to reviewer jZNx**
>
> **2. More broadly, I'm not sure how relevant this notion of convergence is in the non-convex setting. The local geometry of loss landscapes in deep learning can change quite rapidly. One of the effects of this is that certain optimizers can get to regions with small gradients, but they are still far from optimal (and indeed have poor performance). This can be seen even when comparing SGD and Adam in NLP tasks in the large/full batch setting [4]. This is reflected in the loss trajectories which slow down very rapidly for SGD; for full batch training, $\frac{d\mathcal{L}}{dt}$  is proportional to the gradient norm squared. Therefore SGD appears to ``converge'' better and faster according to the definition in this paper, when in reality it is perhaps just optimizing poorly.**
>
> _Response:_ In the context of non-convex losses, a wide assumption that the community takes is that the loss is $L$-smooth. From this $L$ smooth property, one can derive convergence with the popular optimisers. There are many works in literature [2,3,4,5,6,7,8] that have thoroughly studied the convergence of Adam in non-convex settings, but they all prove the convergence with decreasing step sizes. Using a decreasing step size makes it easier to get tighter convergence bounds, but this may violate the true notion of convergence (driving the grad norm of loss to zero). We have clearly explained in our manuscript **(Section 3)** how may the notion of convergence be violated due to decreasing step size.
>
> _We tried to summarize it here, but we guess there are some Latex issues with the open review interface. Kindly refer to Section 3 of our main manuscript_
>
> Also, you have pointed out correctly that **“optimizers can get to regions with small gradients, but they are still far from optimal”**. Our work studies the same effect, and we define the notion of convergence as the small gradient norm value. We theoretically show that  ADAM with our proposed learning rate guarantees convergence. Similar to the most works cited above, we only show convergence and do not provide any theoretical proof that our proposed step-size coupled with ADAM can find the best local minima. However, we provide extensive  empirical evidence that using our proposed step size, one can get to a favourable minima with high accuracy value with various neural network architectures.
>
> **References**
>
> [1] Naichen Shi, Dawei Li, Mingyi Hong, and Ruoyu Sun. Rmsprop converges with proper hyper-parameter. In
> International Conference on Learning Representations, 2020.
>
> [2] Ran Tian and Ankur P Parikh. Amos: An adam-style optimizer with adaptive weight decay towards model-
> oriented scale. arXiv preprint arXiv:2210.11693, 2022.
>
> [3] Xiangyi Chen, Sijia Liu, Ruoyu Sun, and Mingyi Hong. On the convergence of a class of adam-type algorithms
> for non-convex optimization. arXiv preprint arXiv:1808.02941, 2018.
>
> [4] Liangchen Luo, Yuanhao Xiong, Yan Liu, and Xu Sun. Adaptive gradient methods with dynamic bound of
> learning rate. arXiv preprint arXiv:1902.09843, 2019.
>
> [5] Alexandre Défossez, Léon Bottou, Francis Bach, and Nicolas Usunier. A simple convergence proof of adam
> and adagrad. arXiv preprint arXiv:2003.02395, 2020.
>
> [6] Fangyu Zou, Li Shen, Zequn Jie, Weizhong Zhang, and Wei Liu. A sufficient condition for convergences
> of adam and rmsprop. In Proceedings of the IEEE/CVF Conference on computer vision and pattern
> recognition, pp. 11127–11135, 2019.

---

> > ### Author Response · Authors · 2024-09-17
> > **Response to reviewer jZNx**
> >
> > **3. There are also issues with the form of the stepsize. For one, the maximum value of the Hessian (and therefore the local lipschitz constant) can vary wildly for neural networks over initialization space, particularly in the limit of large dimensions or with ReLU activations. Indeed, even for a training trajectory we can observe massive increases in the maximum Hessian eigenvalue. This increasing phenomenon can be even worse in Adam. This could lead to very small values of K  and therefore small stepsize; this suggests that the form of the stepsize is very sensitive to estimation of K. Indeed, there are many situations where conservative estimation of K  can lead to more practical stepsize recommendations than more accurate ones.**
> >
> > _Response:_
> >
> > Our first response is w.r.t to the line **This increasing phenomenon can be even worse in Adam. This could lead to very small values of K  and therefore small stepsize; this suggests that the form of the stepsize is very sensitive to estimation of K**.
> >
> > For a given real valued function $f: \mathbb{R}^d \rightarrow \mathbb{R}$, it is said to be $L$-smooth if:
> >
> > $\forall \textbf{w}_1, \textbf{w}_2 \in \mathbb{R}^d$     $||\nabla f(\textbf{w}_1) - \nabla f(\textbf{w}_2)||_2 \leq L||\textbf{w}_1 - \textbf{w}_2||_2$
> >
> > Now, lets analyse the second order derivate of $f$, let $\textbf{w}_1$ be a point and and let's assume that $\textbf{w}$ be a variable in the $\delta$ neighbourhood of $\textbf{w}_1$, then we can write the second order derivative in the $\delta$ neighbourhood of $\textbf{w}_1$ as:
> >
> > $\nabla^2 f (\textbf{w}) = \frac{\nabla f(\textbf{w}_1) - \nabla f(\textbf{w})}{\textbf{w}_1 - \textbf{w}}$
> >
> > Now, taking 2-norm on both sides:
> >
> > $||\nabla^2 f (\textbf{w})||_2 = ||\frac{\nabla f(w_1) - \nabla f(\textbf{w})}{\textbf{w}_1 - \textbf{w}}||_2$
> >
> > Using Cauchy Schwartz inequality
> >
> > $||\nabla^2 f (\textbf{w})||_2 \leq \frac{||\nabla f(w_1) - \nabla f(\textbf{w})||_2}{||\textbf{w}_1 - \textbf{w}||_2}$
> >
> > $\sigma_{max} \leq \frac{||\nabla f(w_1) - \nabla f(\textbf{w})||_2}{||\textbf{w}_1 - \textbf{w}||_2}$
> >
> > Here, $\sigma_{max}$ is the maximum eigen vale of the hessian of $f$, and hence the Lipschitz smoothness constant for the function $f$. The Lipschitz smoothness constant and max eigenvalue of the hessian share a direct relationship.
> >
> > Our learning rate follows an inverse relationship with the Lipschitz value; hence when the Lipschitz is high, the learning rate becomes low. A large Lipschitz indicates a steep curvature in some directions, which necessitates a lower learning rate to prevent overshooting during optimization. However, when Lipschitz is large, it often suggests that the model's loss landscape has sharp minima or high curvature regions, which can be indicative of overfitting or poor generalization. In such scenarios, the justification for a lower learning rate includes the following points:
> >
> > **1. Stability and Control:** A lower learning rate in high-curvature regions helps in maintaining stability during training. It prevents large updates that could lead the optimizer to jump out of a narrow, potentially optimal region. This stability is crucial, especially when dealing with sharp minima, which are associated with high eigenvalues of the Hessian.
> >
> > **2. Adjusting for High Curvature:** When the Hessian’s maximum eigenvalue is large, it indicates that the loss function is highly sensitive in some directions. A lower learning rate allows the optimizer to adjust carefully, thereby avoiding drastic changes that could worsen generalization by overfitting to the training data.
> >
> > This response continued to the next block due to the character limit. Contd.....

---

> > > ### Author Response · Authors · 2024-09-17
> > > **Response to reviewer jZNx (Continuation of 3)**
> > >
> > > Our response w.r.t the line **Indeed, there are many situations where conservative estimation of $K$  can lead to more practical stepsize recommendations than more accurate ones.**
> > >
> > > _Response:_ **Conservative Estimation of $K$:**  **Algorithm 2**, proposed for estimating the Lipschitz smoothness/constant of the loss, assumes that the probability distribution from which weights are sampled must have a full domain. Under this assumption, the algorithm converges in distribution to the true Lipschitz constant of the network. However, in practice, sampling from regions where the probability density function is very low (close to zero) is not feasible. As a result, **Algorithm 2** typically yields a conservative lower bound on the true Lipschitz smoothness. While asymptotically, under ideal conditions, the algorithm can estimate the true global Lipschitz constant, practical limitations—such as the inability to sample extreme values from distributions like the standard normal—mean that **Algorithm 2** often estimates a conservative local Lipschitz constant. This conservative estimation can be connected to better generalization.
> > >
> > > **Connection to Better Generalization:** When sampling points from a standard normal distribution and using Algorithm 2 to estimate the Lipschitz smoothness of the loss, we implicitly ensure that the model's weights are drawn from a standard normal distribution during training. This approach effectively computes the Lipschitz constant for an $L_2$ regularized loss, consistent with the Maximum A Posteriori (MAP) estimation criteria. Consequently, this process promotes a weight space which is less prone to poor generalization, helping the model avoid overfitting and enhancing its performance on unseen data.

---

> > > > ### Comment · Reviewer_JZNx · 2024-09-17
> > > > **Comment on lipschitz estimation**
> > > >
> > > > I think my overall point still stands; the very conservative estimation of $K$ is still key to getting anything practical out of the results. We know from random matrix theory that spectra of neural networks at initialization concentrate in the high dimensional limit. In contrast, training takes us to parts of the landscape that are non-generic, and therefore not well described by initialization/Gaussian distributions over the weights. The real issue is not even lack of ability to sample from Gaussian tails; it's the non-independence of parameters after optimization.

---

> ### Author Response · Authors · 2024-09-17
> **Response to reviewer jZNx**
>
> **4. The stepsize also depends on $\rho/T$ . In many settings there are a wide range (at least a factor of 10) of $\rho$ which train well, with similar stepsizes. The total training time $T$  may also vary widely during an experimental run. This would suggest that the proposed stepsize is also sensitive to the exact value of $\rho$  and $T$ , and this can also cause deviation away from good stepsizes found by tuning. Another issue with the experiments is that the theoretical learning rate depends on the training horizon $T$, but the paper didn't present empirical evidence that tuning the horizon maintains even the optimization superiority of the theoretical fixed learning rate. Training horizon is an important hyperparameter to tune in practical settings, so understanding this is important to any new methods.**
>
> _Response:_ Thanks for your valuable feedback. Following your suggestions, we have now included two new Sections in our revised manuscript studying the effect of $T$ and $\rho$ on the convergence with our step size: **Section 5.5** for “Tuning T” experiments and Section 5.6 for “Tuning $\rho$”. In our experiments with Tuning $T$, we found that for various values of $T$ (=20,50,100,150) our proposed step-size drives the gradient norm to zero more aggressively as compared to the state-of-the-art schedules and other constant step-sizes, ensuring better convergence.
>
> Since $\rho$ is used solely to prevent the gradient accumulation term $\textbf{V}^{-1/2}$ from exploding due to zero values, setting $\rho$ too high may affect the optimization performance of Adam. From **Figure 16**, we observed a slight overshoot in the gradient norm and loss for higher values of $\rho$. As $\rho$ decreases, the training process tends to become smoother.
>
> _Please refer to the **Section 5.5 and 5.6** for more details and insights._
>
> **5. I have major concerns with the experiments. The first is that the theoretically predicted learning rate is larger than 10−2  (I think this is the case; I could not find the actual value listed anywhere). However, the sweep has maximum learning rate 10−2 . This means that the other settings all have learning rate less than the theoretical stepsize at all steps. This makes the experiments at best, inconclusive.**
>
> _Response:_ The theoretically predicted learning rate is in the order of $10^{-2}$. The actual values of our proposed learning rate are given in every figure of the revised manuscript. In all the experiments which deal with comparison with constant step size, we have now increased our sweep from $10^{-2}$ to $10^{-1}$. Additionally, we conducted a fine search around our proposed step size $\alpha_{ours}$ by varying it by a factor of 2 as per your suggestion. The final comparison is made using the list {10−1, 10−2, 10−3, 10−4, 10−5, $\alpha_{ours}/2$, and $2 \alpha_{ours}$}. Kindly check our modified **Section 5.2 and 5.3**.
>
> From **Figure 2**, we observe that our proposed step size effectively reduces the gradient norm of the loss function. Although the constant learning rate of $10^{-1}$ achieves a slightly lower gradient norm than our method (the difference is minimal and nearly negligible), it shows noticeable overshoots during training. This pattern is also evident in other figures, such as **Figures 4,8,11.** In **Figure 8 and 11**, the overshoots are more pronounced compared to **Figure 2**. This is because, in CNNs, which are more complex models than small fully connected networks, finding local minima is more challenging with such a large learning rate. The learning rate of $10^{-1}$ is relatively high for training neural networks, which can lead to overshooting of local minima or maxima. In contrast, our learning rate decreases both the loss and its gradient norm smoothly during training.

---

> ### Author Response · Authors · 2024-09-17
> **Response to reviewer jZNx**
>
> **6. There is also a question of setting; the mini-CIFAR10 and the mini-MNIST are both very small datasets, and it is unclear what general lessons we can draw from this setup. Additionally, in both of these settings with any reasonable network, one quickly reaches interpolation, and generalization matters far more than optimization/training performance - where, to my eye, the other methods have a distinct advantage. Additionally, SGD is known to generalize better than Adam on CIFAR10 scale datasets using architectures like ResNet.**
>
> _Response:_ Plots for experiments involving the full MNIST and CIFAR-10 datasets are provided in **Section 5.3** of our main manuscript. In this work, we have proven the convergence of both deterministic and stochastic versions of Adam. For deterministic Adam, we require a dataset that can pass through the network entirely without batching; therefore, mini versions of MNIST and CIFAR-10 were created to empirically validate the convergence of deterministic Adam with our proposed step size **(Theorem 1)**.
>
> Both deterministic and stochastic Adam demonstrate superior convergence with our proposed step size, as shown in the manuscript. Additionally, they exhibit better generalization, as indicated by the test accuracy versus epochs plots in each experiment. Note that the y-axis of these graphs does not display accuracy in percentage. To convert the values into percentages, multiply them by 100. We have added these instructions to our revised manuscript to prevent any confusion. As illustrated in the accuracy graphs, our learning rate (along with others) converges to high accuracy levels.
>
> **7. Overall I also don't understand the point of the comparison of learning rate schedules to the theoretically chosen fixed learning rate. In all major applications, Adam is paired with either a decaying learning rate schedule or an averaging scheme; constant learning rate schedules just don't work well on their own.**
>
> _Response:_ Kindly refer to **Section 3** of our main manuscript. A detailed explanation on _why choose constant step-size_ is given, which is purely from the point of view of **Convergence of Adam**. Again, we tried to summarize it here, but we guess there are some Latex issues with the open review interface. Kindly refer to **Section 3** of our main manuscript
>
> **Major Changes Requested**
>
> **1. I would like to see a more precise definition for the theorem so I can try to understand it better.**: Typos from the both main theorems are removed.  To summarize both the theorems $\rightarrow$ _When we discuss ensuring convergence in Theorems 1 and 2, we refer specifically to the convergence where the 2-norm of the gradient of the loss function approaches zero, which is a common measure of convergence, although there are other definitions. In this work, we derived a constant step size that guarantees convergence of Adam by driving the gradient norm of the loss to zero after a certain number of iterations, which is denoted by $T(\beta_1,\rho)$ in our manuscript_
>
> **2. All experimental sweeps should include values larger than, and have grid spacing of factor of 2 or 3 (at least near the optimal learning rates).** : Addressed in the above response. Kindly refer to the modified **Section 5.2 and 5.3**.
>
> **3. Additional experiments with varying T should be conducted and put in the main text.**: Addressed in the above response. Kindly refer to the newly added section in the main manuscript, **Section 5.5**.
>
> **Minor Changes Requested**
>
> **1. In the abstract, remove the (i), (ii), etc, they don't add much.**: Addressed. Kindly check our modified abstract.
>
> **2. In Equation 1, what is v?**: Addressed. The **v** is Adam's second-order momentum parameter (Check **Algorithm 1**). We have modified **Section 1** now; we have defined the Adam algorithm first and then the paragraph where **v** was discussed.
>
> **3. If vectors are boldface, matrices should be as well.**: Addressed.
>
> **4. Some of the references are poorly formatted; likely a bibliography settings issue.**: Fixed.

---

> > ### Comment · Reviewer_JZNx · 2024-09-17
> > **Question on Figure 1**
> >
> > What is the base learning rate used for the scheduled methods here?

---

> > > ### Author Response · Authors · 2024-09-19
> > > **Response for base learning rate**
> > >
> > > For various schedulers, we fine-tuned their hyperparameters to identify configurations where they exhibited the best performance and retained those values for the final comparison with our step size. These choices are made to provide a fair comparison with our constant step size approach, as overly rapid decay rates may not effectively minimise the gradient norm of the loss function. We choose a base learning rate of $10^{-2}$.

---

> > ### Comment · Reviewer_JZNx · 2024-09-17
> > **Overall response to changes**
> >
> > I thank the author for their changes, the presentation of the theorems is improved and the overall experimental section has been improved as well.
> >
> > My overall points still remain however; the sensitivity of the functional form of the learning rate to $\rho$ and $T$, the need for a poor/conservative approximation of $K$ to get an interesting learning rate, the questionable/impractical notion of convergence, and the fact that constant learning rate schedules (no matter how carefully chosen) are not useful in practical settings.

---

> > > ### Author Response · Authors · 2024-09-19
> > > **Replying to "Overall response to changes" by reviewer jZNx**
> > >
> > > **1.  The questionable/impractical notion of convergence**
> > >
> > > _Response:_ A widely used notion of convergence in gradient-based methods is when the gradient of the objective function with respect to the parameters approaches zero, meaning the updates become smaller, and the algorithm approaches a stationary point (where the gradient is zero).
> > >
> > > Mathematically, this is expressed as:
> > >
> > > $\lim_{t \to \infty} ||\nabla_{\textbf{w}} \mathcal{L}(\textbf{w}_t)||_2 = 0$ or
> > >
> > > $||\nabla_{\textbf{w}} \mathcal{L}(\textbf{w}_t)||_2 < \epsilon$ for some $t \geq \hat{T}$ where $\hat{T}$ is a natural number.
> > >
> > > This means that, as the number of iterations $t$ increases, the gradient norm gradually approaches zero, implying that the optimization process is nearing a stationary point.
> > >
> > > In much of the literature, convergence is often defined as:
> > >
> > > **Definition:** In iterative gradient-based optimization, convergence refers to the behaviour of the objective function value, the parameters, or the gradient itself, with the most common condition being that the gradient norm approaches zero. This indicates that the optimization algorithm is reaching a stationary point, where the gradient vanishes. This notion of convergence is widely used in many previous works [1,2,3,4,5,6].
> > >
> > > Hence, in all these works, the primary focus was to provide rigorous theoretical bounds for the convergence of stochastic optimizers. As mentioned earlier, convergence here implies that the gradient norm of the objective function becomes zero, indicating that the algorithm reaches a stationary point (which could be a local minimum or a point of inflection). These research efforts aim to prove convergence to a stationary point, not necessarily to a favorable minimum.
> > >
> > > However, extensive _empirical_ evidence suggests that while converging, optimizers like Adam and RMSProp often reach a local minimum. This maybe due to the fact that they use information from past gradients, and it allows them to take a trajectory that will eventually lead to a better local minima than vanilla SGD. Hence, adaptive optimisers are a hot topic in deep learning. In our paper, we have conducted similar experiments, demonstrating that Adam with a constant step size _empirically_ converges to a good local minimum and provide rigorous _theoretical_ analysis only for the convergence part.
> > >
> > > We hope that addresses your concern. Please let us know if you are expecting any other type of explanation.
> > >
> > >
> > > **References**
> > >
> > > [1] Naichen Shi, Dawei Li, Mingyi Hong, and Ruoyu Sun. Rmsprop converges with a proper hyper-parameter. In International Conference on Learning Representations, 2020.
> > >
> > > [2] Ran Tian and Ankur P Parikh. Amos: An adam-style optimizer with adaptive weight decay towards model- oriented scale. arXiv preprint arXiv:2210.11693, 2022.
> > >
> > > [3] Xiangyi Chen, Sijia Liu, Ruoyu Sun, and Mingyi Hong. On the convergence of a class of adam-type algorithms for non-convex optimization. arXiv preprint arXiv:1808.02941, 2018.
> > >
> > > [4] Liangchen Luo, Yuanhao Xiong, Yan Liu, and Xu Sun. Adaptive gradient methods with dynamic bound of learning rate. arXiv preprint arXiv:1902.09843, 2019.
> > >
> > > [5] Alexandre Défossez, Léon Bottou, Francis Bach, and Nicolas Usunier. A simple convergence proof of adam and adagrad. arXiv preprint arXiv:2003.02395, 2020.
> > >
> > > [6] Fangyu Zou, Li Shen, Zequn Jie, Weizhong Zhang, and Wei Liu. A sufficient condition for convergences of adam and rmsprop. In Proceedings of the IEEE/CVF Conference on computer vision and pattern recognition, pp. 11127–11135, 2019.

---

> ### Author Response · Authors · 2024-09-19
> **Replying to "Overall response to changes" by reviewer jZNx (2)**
>
> **2. The sensitivity of the functional form of the learning rate to $\rho$  and $T$.**
>
> _Response:_ As per your suggestion, we have already provided the sensitivity analysis of  $\rho$ and $T$ on ADAM with our proposed step size. Kindly refer to **Sections 5.5 and 5.6** in the revised manuscript for in-depth analysis. We again provide the summary of our experiments below for your convenience. We hope that addresses your concern. Please let us know if you are expecting any other analysis from us.
>
> **Part 1: Sensitivity of the functional form of the learning rate to $\rho$**
>
> The Adam update rule can be written as follows:
>
> $\textbf{w}_{t+1} = \textbf{w}_t - \alpha_t \frac{\hat{m}_t}{\sqrt{\hat{v}_t + \rho}}$
>
> where $\hat{m}_t$ is the first-moment estimate, $\hat{v}_t$ is the second-moment estimate (variance), and $\rho$ is a small positive constant added to prevent division by zero or very small values.
>
> The parameter $\rho$ is introduced to ensure that the second-order momentum term$\sqrt{\hat{v}_t}$ does not explode. Thus, $\rho$ is typically chosen to be a very small positive number. _This is not a parameter of our learning rate, but it is a crucial hyper-parameter for the Adam algorithm_. For example, the default value in the PyTorch package is $10^{-8}$. $\rho$ is one of the key parameters of Adam, along with $\beta_1$ and $\beta_2$, which control the behaviour of the optimizer as it navigates the loss surface. Deviating significantly from the default value/range of $\rho$ can negatively affect Adam’s performance, altering the trajectory it follows during optimization. Even with state-of-the-art learning rate schedulers, if parameters like $\rho$, $\beta_1$, and $\beta_2$ deviate too much from their default values or typical ranges, optimizers like Adam or RMSProp may not perform well, and in some cases, they may not even converge.
>
> _Impact of $\rho$ in our experiments:_ In our experiments, we observe that increasing $\rho$ can lead to an overshoot in the gradient norm. This occurs because a higher value of $\rho$ interferes with the second-order _exponential moving average_ (EMA), preventing Adam from navigating the loss landscape effectively. This highlights the importance of keeping $\rho$ within a reasonable range to ensure stable and effective optimization.
>
> **Part 2: Sensitivity of the functional form of the learning rate to T**
>
> $T$ is a hyperparameter _in our proposed step size, not a parameter of Adam_. As instructed by you, we have provided detailed experiments on varying $T$ in **Section 5.5**. In all the experiments of **Section 5.5**, with all the values of $T \in (20,50,100,150)$, our proposed step size converges faster with good accuracy in the classification task.
> Note that $T$ is a step size parameter, and it is not a core parameter for the Adam optimiser. With core parameters tampered with, even the SOTA schedulers cannot get Adam working properly. Hence, with an optimal and widely accepted set of core parameters of Adam, our proposed step size makes Adam converge as compared to other SOTA schedulers.

---

> > ### Author Response · Authors · 2024-09-19
> > **Replying to "Overall response to changes" by reviewer jZNx (3)**
> >
> > **3. The need for a poor/conservative approximation of K to get an interesting learning rate**
> >
> > _Response:_ We will reply to your latest comment on estimation of $K$ **the very conservative estimation of K  is still key to getting anything practical out of the results. We know from random matrix theory that spectra of neural networks at initialization concentrate in the high dimensional limit. In contrast, training takes us to parts of the landscape that are non-generic, and therefore not well described by initialization/Gaussian distributions over the weights. The real issue is not even lack of ability to sample from Gaussian tails; it's the non-independence of parameters after optimization.**
> >
> > We agree with your point that **We know from random matrix theory that spectra of neural networks at initialization concentrate in the high dimensional limit. In contrast, training takes us to parts of the landscape that are non-generic, and therefore not well described by initialization/Gaussian distributions over the weights.** We will try to address it as follows:
> >
> > Consider a true risk minimization problem $\min_{\textbf{w} \in \mathbb{R}^d} \mathbb{E}_{x \sim \mathcal{P}}\mathcal{L}(\textbf{w},\textbf{x})$. Suppose we use Adam to solve this optimization problem. The weights at every iteration can be drawn from any point in the large $\mathbb{R}^d$ space by Adam iterates. Now, let $K$ be the Lipschitz constant of the loss function, keeping in mind that the full $\mathbb{R}^d$ is the domain. We refer to this as the global Lipschitz constant or the true Lipschitz constant of the function.
> >
> > Next, consider a regularized risk minimization problem $\min_{\textbf{w} \in \mathbb{R}^d} \mathbb{E}_{x \sim \mathcal{P}}\mathcal{L}(\textbf{w},\textbf{x}) + \lambda ||\textbf{w}||^2$, lets call it $Eq. (2)$. According to _Maximum A Posteriori_ (MAP) estimation theory, this optimization is equivalent to assuming that the weights are drawn from a Gaussian distribution _at every iteration_, implying a Gaussian prior over the network weights. In this case, the effective domain of the optimization is reduced, as the weights are constrained by the Gaussian prior, which restricts the domain to a subset smaller than the full $\mathbb{R}^d$ space.
> >
> > If we estimate the Lipschitz constant in this restricted domain and denote it as $K'$, we will have $K' \leq K$, making $K'$ a conservative estimation of the true global Lipschitz constant $K$. Our **Algorithm 2**, when applied to weights sampled from a Gaussian distribution, asymptotically approaches a tighter lower bound on $K$. Therefore, **Algorithm 2** estimates a conservative Lipschitz bound in this scenario. For **N**) iterations, we keep sampling weights from a Gaussian distribution, implying that during optimization, the model weights are drawn from this distribution, similar to the setup in Equation (2). As we keep changing distribution, the prior also changes, but the optimisation problem remains the same as (2).
> >
> > Note: Say, ${w_1,...,w_N}$ be the set of weights the optimiser attains at every iteration for solving $Eq. (2)$ and let ${w_1’,...,w_N’}$ be the set of weights we sample for **N** iteration to estimate Lipschitz, these two sets may not be the same, but as we provide an asymptotical proof to our **Algorithm 2** (**Theorem 4**), we tend to find the tight Lipschitz constant. Practically it is achievable by increasing the number of iterations.
> >
> > We hope that addresses your concern.
> >
> > **4. And the fact that constant learning rate schedules (no matter how carefully chosen) are not useful in practical settings.**
> >
> > _Response:_ A detailed analysis of why we choose a constant step size over schedulers is provided in **Section 3** of our main manuscript. We kindly request you to review it. In that section, we present sufficient theoretical evidence demonstrating how schedulers can slow down the convergence of Adam iterates. Additionally, we provide empirical evidence through graphs showing that, when using schedulers, the gradient norm decreases less aggressively compared to our proposed constant step size.
> >
> > This analysis **focuses exclusively on the convergence aspect**, demonstrating that better convergence can be achieved with a constant step size compared to schedulers. It suggests that a constant step size can be more effective in driving convergence than schedulers. However, it does not explore which strategy—schedulers or constant step size—is more effective in reaching a **good local minimum** (Convergence analysis and finding a good local minima are two different sets of problems).  In line with previous work (same cited above), we provide _theoretical evidence_ of Adam's convergence with our step size and support it with _extensive empirical_ evidence showing that it converges to local minima.

---

### Review · Reviewer_Aq3G · 2024-09-02

**Summary Of Contributions:**

This article analyzes the convergence of Adam algorithm with a constant step size in non-convex optimization. The main theoretical result shows that if the step size is chosen properly, the algorithm can find a critical point of the loss function. The proposed step size in theory is also evaluated in practice, based on a method to estimate Lipschitz smoothness of the loss function, showing empirically the importance of the use of a constant step size.

**Audience:**

Yes

**Claims And Evidence:**

No

**Requested Changes:**

- Did you discuss how the proposed constant step size in Section 4 is used in practice in Section 5 ?
- Specify the distribution W in Algorithm 2.
- Check the statement of Theorem 1: ... for some $t \geq T(\beta_1,\rho)$ ... are you trying to say for some $T \geq  T(\beta_1,\rho)$ ?
- Make the notation consistent in Section 2 and 3, by either using $f$ or $L$ as the loss, same for the $x$ and $w$. This is particularly confusing in Theorem 2.

**Strengths And Weaknesses:**

Strength:
- the article studies an interesting scenario of the Adam algorithm, different to existing works on decaying learning rate.
- the theoretical result seems to be novel with under reasonable assumptions of the loss function.

Weakness:
- Significant: the constant step size choice depends on the minimal loss value, which is in practice unknown.
It is unclear how this could be used evaluated numerically.
- Soundness: the method Algorithm 2 assumes a domain W to define a probability distribution, it is unclear what this distribution is if the full domain is unbounded.
- Writing: The notations are not consistent in Section 2 and section 3, some statement of main results seems to contain typo.

---

> ### Author Response · Authors · 2024-09-17
> **Response to reviewer Aq3G**
>
> Thank you for reviewing our paper and providing insightful comments and feedback. We have tried to address your concerns by providing explanations below and revising our paper.
>
> **1. The constant step size choice depends on the minimal loss value, which is in practice, unknown. It is unclear how this could be used and evaluated numerically.**
>
> _Response:_  In all our experiments, the loss function used is cross-entropy loss, which has a minimum value of 0. Other well-known loss functions in the literature, such as MSE, MAE, f-divergence-based loss, and InfoNCE loss, also have a minimum value of 0. We assume that the true risk minimization problem is solved accurately. While exploring other complex loss functions is beyond the scope of this study, it will be addressed in future work. We thank the reviewer for highlighting this point.
>
> **Loss Function Design:** When designing a loss function for minimization, it is essential to ensure the following criteria are met:
>
> 1. **Bounded from Below:** The loss function should be bounded from below to prevent the minimization process from collapsing towards negative infinity.
>
> 2. **Differentiability:** The loss function should be at least once differentiable, allowing the use of first-order iterative gradient methods to find the minima.
>
> 3. **Lipschitz Smoothness:** The loss function should be \(L\)-smooth, ensuring a smooth loss surface that facilitates navigation by iterative methods.
>
> From the first point, having access to the minimum value of the loss function is advantageous. If the exact minimum is not accessible, a lower bound can always be used as a reference.
>
> **2. The method Algorithm 2 assumes a domain W to define a probability distribution, it is unclear what this distribution is if the full domain is unbounded.**
>
> _Response:_ **Algorithm 2** assumes that $\mathcal{W}$ is a full-domain distribution, but it does not assume that $\mathcal{W}$ is both full-domain and unbounded. Examples of full-domain distributions include Gaussian and Laplacian distributions, whose domain ranges from $(- \infty, + \infty)$. Since the estimation of our Lipschitz constant depends on $\mathcal{W}$, we presented a sensitivity analysis in **Appendix Section C.5**, that involves estimating Lipschitz smoothness using various distributions (both bounded and unbounded domains) and applying our proposed step size with the obtained Lipschitz values. These experiments empirically show that our proposed learning rate is not highly sensitive to the choice of $\mathcal{W}$.
>
> **3. The notations are not consistent in Section 2 and section 3, some statement of main results seems to contain typo.**
>
> _Response:_ We have addressed this comment. All notations are now consistent in our revised manuscript. All optimizing variables are now denoted by $\textbf{w}$. In the definitions, however, we use $f$ to indicate that these definitions apply to a broader and more generic class of functions. For loss functions, we denote them by $\mathcal{L}$ in theorems when it is necessary to specify, particularly if the theorem is tailored explicitly for loss functions. **Theorem 2** is now fixed. Thank you for your suggestions to improve our manuscript.
>
> **4. Did you discuss how the proposed constant step size in Section 4 is used in practice in Section 5 ?**
>
> _Response:_ Yes, in the revised version, our proposed step size is denoted by $\alpha_{ours}$ and explicitly stated in **Section 4.2**. We use the same notation in **Section 5**.
>
> **5. Specify the distribution $\mathcal{W}$ in Algorithm 2.**
>
> _Response:_ In all our main text experiments, we have used standard normal distribution. Please check **Appendix Section C.5** for more experiments with various other distributions.
>
> **6. Check the statement of Theorem 1.**
> _Response:_  It is $t \geq T(\beta,\rho)$. We have rewritten **Theorem 1 and 2** to avoid any confusion. In this work, we derived a constant step size that guarantees the convergence of Adam by driving the gradient norm of the loss to zero after a certain number of iterations, which is $T(\beta,\rho)$.
>
> **6. Make the notation consistent in Section 2 and 3.**
>
> _Response:_  All notations are consistent in our revised manuscript. All optimizing variables are now denoted by $\textbf{w}$. In the definitions, however, we use $f$ to indicate that these definitions apply to a broader and more generic class of functions. For loss functions, we denote them by $\mathcal{L}$ in theorems when it is necessary to specify, particularly if the theorem is tailored explicitly for loss functions. **Theorem 2** is now fixed. Thank you for your suggestions to improve our manuscript.

---

> > ### Comment · Reviewer_Aq3G · 2024-10-01
> > **some comments**
> >
> > Dear authors,
> > Thanks for your detailed reply, the quality of the article is indeed improved. I still have questions about some technical aspects:
> > - How do you address the case if min L(w) is non-zero, but we know a prior that L(w) >= 0 for any w in the full domain. In other words, do Theorem 1 and 2 still work if L(w*) is replaced by a lower bound?
> > - Can you provide a proof of Theorem 4 ? I did not find your appendix.
> >
> > regards,
> > reviewer

---

> > > ### Author Response · Authors · 2024-10-02
> > > **Reply to reviewer A13G on latest comment**
> > >
> > > **1. How do you address the case if min L(w) is non-zero, but we know a prior that $\mathcal{L}(w) \geq 0$ for any $w$ in the full domain. In other words, do Theorem 1 and 2 still work if $\mathcal{L}(w^{*})$ is replaced by a lower bound?**
> > >
> > > _Response:_ Our proposed step size is:
> > >
> > > \alpha_{ours} = \sqrt{\frac{2(L(w_{0}) - L(w^{*}))}{K.T}}
> > >
> > > If the loss min value is zero, we can remove the $\mathcal{L}(w^*)$ term. Otherwise, we need to account for the minimum value of the loss. As previously discussed, most common loss functions have zero as their minimum value. A non-zero minimum value can occur if a shifted version of the loss is used.
> > >
> > >
> > > **Theorems 1 and 2** state that as the number of iterations tends to infinity, the gradient norm converges to zero. This happens in a minimization problem for two reasons: either the iterator reaches a point of inflection or it reaches a local minimum. If the minimum does not exist, the same property holds for the infimum, which is the tightest lower bound of the function.
> > >
> > >
> > > To answer your question: Yes, Theorems 1 and 2 still hold if $\mathcal{L}(w^*)$ is replaced by the **tightest lower bound** (infimum). This is because the gradient norm converges to zero at the infimum as well. However, if you choose any other lower bound (i.e., a loose bound), it is not possible to make claims about the gradient behaviour. In such a case, you would need to adjust the hyperparameter $\epsilon$ in the theorem or establish a lower bound on the gradient norm itself. It may be necessary to define a new notion of convergence, where the iterator is considered to converge if the gradient norm lies within the range $[a, b]$ for some $t \geq T(.)$, then **Theorem 1 and 2** will be modified of the new notion of convergence. But, for practicality, it is always advised to choose a loss function which has a well defined minimum value or infimum.
> > >
> > > It is generally easier to find the minimum value of a function when the functions are _real-valued_, even though finding the corresponding **minimizer** can still be challenging as it depends on what type of domain the fucntion have.
> > >
> > > We acknowledge that handling more complex and intricate loss functions is an important direction, and we will consider this in our future work. We sincerely thank the reviewers for their valuable feedback and insightful suggestions that have helped us improve the quality of this manuscript.
> > >
> > > **2. Can you provide a proof of Theorem 4 ? I did not find your appendix.**
> > >
> > > _Response:_ We apologize for the typo in the Appendix where Theorem 4 was mistakenly referenced as Theorem 3. We have corrected it and uploaded the modified Appendix. Thank you for bringing this to our attention.

---

### Review · Reviewer_MXan · 2024-09-04

**Summary Of Contributions:**

This paper studies the convergence of the Adam optimizer with a fixed step size. For both deterministic and stochastic Adam, the authors derive the conditions for step size that ensure convergence. The paper provides an exact formulation for the step size as a function of the initial and optimal loss, the smoothness of the loss function, the gradient norm, and the total number of optimization iterations. Additionally, it introduces a method to approximate the Lipschitz continuity of the loss function by evaluating the Hessian spectral norm of randomly sampled weights. Experiments on MNIST and CIFAR-10 with various simple model architectures and baseline learning rate configurations show that the derived fixed learning rate schedule can benefit convergence in these settings.

**Audience:**

Yes

**Claims And Evidence:**

No

**Requested Changes:**

**Presentation:**

Please review all in-text citations for proper formatting, as many are currently incorrect.

**Evaluations:**

For the Lipschitzness approximation, one suggestion is to empirically demonstrate the approximated value versus N (number of iterations) to show increasing accuracy as N increases.

**Clarification:**

How exactly is $\mathcal{W}$ sampled in Algorithm 2?

What exactly is the value of "Ours" in each figures in Section 5? Are all learning rate "in the order of $10^{-2}$" (bottom of page 8)? It would be helpful to directly state it directly in the figures.

(Assuming ours is $10^{-2}$) For Figure 2, it is interesting to see that the grad norm, loss, and test accuracy are in between results with 0.01 and 0.001. One would assume that it should be in between the two curves. Why is it the case?

Can the authors discuss the observation of potentially poorer generalization performance despite faster convergence in loss and grad norm plots in Figure 1. Also please consider elaborating on this in other experimental settings.

**Strengths And Weaknesses:**

**Strength:**

The focus on Adam with a constant learning rate offers valuable insights into the algorithm, complementing existing studies (B1) and (B2).

The paper provides an exact step size formulation and an empirically usable approximate learning rate.

The experimental setup is comprehensively detailed across all settings.

The benefits of Adam with a constant learning rate are empirically demonstrated, showing faster convergence compared to various learning rate schedules and other standard constant rates.

**Weakness:**

While the paper proposes a method for estimating the Lipschitzness of the loss function, no empirical results validate its efficacy, nor are comparisons made with other methods, such as those in Gouk et al. (2021).

Additionally, the efficiency claim in approximating the Lipschitz constant should be considered cautiously, as there are no runtime comparisons with other approximation methods.

---

> ### Author Response · Authors · 2024-09-17
> **Author response to reviewer MAxn**
>
> Thank you for your valuable comments and thorough suggestions. We have addressed all identified weaknesses and revised the manuscript, highlighting the modifications in “dark purple” (In Appendix, Additional Experiments Section C.1) according to the required changes. Please review the updated manuscript for detailed changes.
>
> **1. While the paper proposes a method for estimating the Lipschitzness of the loss function, no empirical results validate its efficacy, nor are comparisons made with other methods, such as those in Gouk et al. (2021). Additionally, the efficiency claim in approximating the Lipschitz constant should be considered cautiously, as there are no runtime comparisons with other approximation methods.**
>
> $\textit{Response:}$ We have added a fully separate section on the analysis and comparison of our Lipschitz estimation method with other efficient Lipschitz estimation methods in our Appendix (Section C.1). We have made runtime comparisons with methods like LipSDP Neuron [1] and LipSDP Layer [1] which claim to estimate tight upper bounds on the Lipschitz constant of neural networks.
>
> **Table 1:** Runtime comparison (in seconds $\downarrow$) of LipSDP Neuron, LipSDP Layer, and Our Method across different networks and epochs. Here, epochs indicate the training iterations required to obtain the weights used to compute the Lipschitz constant with LipSDP Neuron and Layer. For our method, epochs refer to the number of weight samples from a distribution to compute the Lipschitz constant, corresponding to the parameter \($N$\) in Theorem 4. The best (lowest) runtime is given in **bold**.
>
> | **Epochs** | **Network**          | **LipSDP Neuron** | **LipSDP Layer** | **Ours**   |
> |:----------:|:--------------------:|:-----------------:|:----------------:|:----------:|
> | 20         | 1 layer 100 neuron   |      42.74        |      40.61       |  **38.02** |
> |            | 3 layer 100 neuron   |      51.26        |      41.38       |    **38.55**   |
> |            | 5 layer 100 neuron   |      58.22        |      42.44       |    **38.41**   |
> | 50         | 1 layer 100 neuron   |      99.25        |      97.87       |    **91.27**   |
> |            | 3 layer 100 neuron   |     107.91        |      98.18       |    **91.43**   |
> |            | 5 layer 100 neuron   |     115.44        |      98.52       |    **91.68**   |
> | 100        | 1 layer 100 neuron   |     198.28        |     197.25       | **180.54** |
> |            | 3 layer 100 neuron   |     207.52        |     197.82       |   **181.28**   |
> |            | 5 layer 100 neuron   |     214.43        |     198.39       |   **181.84**   |
>
> As we can see, the time required to estimate the Lipschitz constant increases with the number of layers in the network for the LipSDP Neuron algorithm. In contrast, the time remains constant for both the LipSDP Layer and our method, regardless of the number of layers. Furthermore, our method outperforms both LipSDP variants in terms of runtime, demonstrating its ability to estimate the Lipschitz constant more quickly.
>
> Additionally, Theorem 4 establishes that our method provides a tight lower bound for the Lipschitz constant, and its effectiveness is empirically validated in Figure 17 (a) of the Appendix (Section C.1).
>
> **Reference**
> [1] _Fazlyab et. al._ "Efficient and Accurate Estimation of Lipschitz Constants for Deep Neural Networks", NeurIPS 2019.

---

> ### Author Response · Authors · 2024-09-17
> **Author response to reviewer MAxn**
>
> **2. For the Lipschitzness approximation, one suggestion is to empirically demonstrate the approximated value versus $N$ (number of iterations) to show increasing accuracy as $N$ increases.**
>
> _Response:_ Our proposed learning rate depends on the Lipschitz smoothness of the loss function. We claim that with our learning rate, Adam converges faster than various state-of-the-art schedulers, as motivated in Section 3 of the main text. It also outperforms many commonly used constant step sizes in neural network training, demonstrating its effectiveness and suggesting that our step size can be a quick alternative to line searches for the optimal learning rate. While Lipschitz's smoothness is crucial for estimating our learning rate, we do not claim that it necessarily leads to superior accuracy. Although accuracy (or other relevant metrics) ultimately measures a model's performance, we provide guarantees on Adam's convergence with our proposed step size. _In all our experiments, test accuracy plots show that our proposed step size not only ensures convergence but also finds a favourable local minimum, resulting in higher accuracies. Sufficient empirical evidence on the accuracy is presented in the main text and appendix across various neural network architectures. We will try to build a strong theory that connects the convergence of Adam with finding the best local minima in our next work._
>
> Following your suggestion, we have provided a graph **[Appendix Section C.1, Fig 17(b)]** illustrating the behaviour of the gradient norm of the loss with varying \($N$\). It is evident that with smaller values of \($N$\), the plot shows an overshoot. However, as \($N$\) increases, the training process becomes smoother. This is because our learning rate and Lipschitz smoothness are inversely related, and the Lipschitz smoothness is approximated using a max operator **(Algorithm 2)**. With fewer iterations, the Lipschitz constant may be underestimated (low value), leading to a higher learning rate and an increased likelihood of overshooting.
>
> **3. How exactly $\(\mathcal{W}\)$ is sampled?**
>
> _Response:_ In all our experiments, points from $\(\mathcal{W}\)$ are sampled randomly (random sampling).
>
> **4. What exactly is the value of "Ours" in each figures in Section 5? Are all learning rate "in the order of $10^{-2}$ (bottom of page 8)? It would be helpful to directly state it directly in the figures.**
>
> _Response:_ No, all learning rates are not in order of $10^{-2}$. In all the graphs, we have now explicitly stated our learning rate in the revised version of our paper.
>
> **5. Can the authors discuss the observation of potentially poorer generalization performance despite faster convergence in loss and grad norm plots in Figure 1. Also please consider elaborating on this in other experimental settings.**
>
> _Response:_ Thanks for pointing it out. We had missed providing an important detail. The y-axis does not display accuracy in percentage. To convert the values into percentages, multiply them by 100. We have included this instruction in our revised main text to avoid any confusion. Consequently, as shown in the accuracy graphs, our learning rate (along with others) converges to high accuracy levels. We again apologize for missing this important point in our original manuscript.
>
> **6. (Assuming ours is 10^-2) For Figure 2, it is interesting to see that the grad norm, loss, and test accuracy are in between results with 0.01 and 0.001. One would assume that it should be in between the two curves. Why is this the case?**
>
> _Response:_ We believe this is due to the network architecture. Most of our experiments are conducted on a linear architecture, and involve the classification of the MNIST dataset, which consists of image data. Given the fully connected and linear nature of the architecture, and the fact that MNIST data is not linearly separable, the loss does not decrease to very low values for linear networks. However, as shown in **Figures 8 and 11** (MNIST with LeNet and VGG-9), the loss value falls below $10^{-4}$, indicating that these CNN architectures are better suited to understanding and classifying MNIST data compared to linear models.

---

> > ### Comment · Reviewer_MXan · 2024-09-24
> >
> > Thank you for the detailed response.
> >
> > The question I was referring to regarding the accuracy plot in Figure 1 is that the red curve (ours) shows the lowest test accuracy, even when multiplied by 100.

---

> > > ### Author Response · Authors · 2024-09-28
> > > **Reply to reviewer MXan**
> > >
> > > **Comment:** The question I was referring to regarding the accuracy plot in Figure 1 is that the red curve (ours) shows the lowest test accuracy, even when multiplied by 100.
> > >
> > > _Response:_ Thank you for your feedback. While it’s true that in **Figure 1**, our learning rate achieves the lowest accuracy among all schedulers, the difference is very minimal—which is clearly visible in the figure. In other figures, one can see that our accuracy matches with the best accuracies.
> > >
> > > | **Method**                | **Theoretical convergence bounds** | **Empirical convergence, driving gradient norm to zero** | **Good local minima (empirical evidence)** |
> > > |------------------------------|------------------------------------|---------------------------------------------------------|--------------------------------------------|
> > > | SOTA Schedulers [1,2,3,4,5,6] | ✓                                  | ✗                                                       | ✓                                          |
> > > | An array of constant step size | --                                | Requires grid search over possible fixed LR's, drives gradient norm towards zero but not as aggressively as our proposed step size            | Depends on selected step size              |
> > > | Our Proposed Step Size        | ✓                                  | ✓ (No grid search required, plug and train. Drives gradient norm of loss towards zero aggressively)                                                      | ✓ (Backed by extensive empirical evidence in this manuscript)                                         |
> > >
> > >
> > > From the above table, we argue that our proposed step size guarantees theoretical convergence, drives the gradient norm of the loss function more aggressively towards zero (as shown in our empirical analysis of convergence), and consistently finds suitable local minima, as demonstrated through extensive experiments. Hence, we are the first to provide a comprehensive analysis of Adam's convergence, offering both theoretical insights and empirical evidence that demonstrate its effectiveness in reaching good local minima.
> > > We argue that state-of-the-art schedulers, which typically employ non-increasing step sizes, can hinder Adam’s convergence. This is why we advocate for a constant step size to achieve better convergence. Our empirical results, as shown in all figures, consistently demonstrate that our proposed step size reduces the gradient norm of the loss function more aggressively compared to other state-of-the-art schedulers, leading to faster and better convergence. Moreover, our step size not only minimizes the gradient norm (whether at a local minimum or an inflection point), but also converges to a favorable local minimum, delivering competitive accuracies. Previous work [1,2,3,4,5,6] has also primarily focused on providing convergence guarantees, similar to ours, but with minimal empirical evidence of achieving improved metric scores.
> > > In summary, we present a constant learning rate that is easy to implement, ensures faster convergence, and achieves accuracy in classification tasks that is comparable to or even better than state-of-the-art schedulers.
> > >
> > > We would like to once again thank the reviewers for their valuable feedback, which has greatly helped in improving our manuscript.
> > >
> > >
> > >
> > > **References**
> > >
> > > [1] Naichen Shi, Dawei Li, Mingyi Hong, and Ruoyu Sun. Rmsprop converges with proper hyper-parameter. In International Conference on Learning Representations, 2020.
> > >
> > > [2] Ran Tian and Ankur P Parikh. Amos: An adam-style optimizer with adaptive weight decay towards model- oriented scale. arXiv preprint arXiv:2210.11693, 2022.
> > >
> > > [3] Xiangyi Chen, Sijia Liu, Ruoyu Sun, and Mingyi Hong. On the convergence of a class of adam-type algorithms for non-convex optimization. arXiv preprint arXiv:1808.02941, 2018.
> > >
> > > [4] Liangchen Luo, Yuanhao Xiong, Yan Liu, and Xu Sun. Adaptive gradient methods with dynamic bound of learning rate. arXiv preprint arXiv:1902.09843, 2019.
> > >
> > > [5] Alexandre Défossez, Léon Bottou, Francis Bach, and Nicolas Usunier. A simple convergence proof of adam and adagrad. arXiv preprint arXiv:2003.02395, 2020.
> > >
> > > [6] Fangyu Zou, Li Shen, Zequn Jie, Weizhong Zhang, and Wei Liu. A sufficient condition for convergences of adam and rmsprop. In Proceedings of the IEEE/CVF Conference on computer vision and pattern recognition, pp. 11127–11135, 2019.

---

### Decision · Action_Editor_n7vP · 2024-10-11

**Recommendation:** Reject

**Comment:**

This paper provides a theoretical and empirical analysis of the convergence behavior of the Adam optimizer with a fixed constant step size in non-convex settings. The authors propose a specific constant step size for Adam, derived from the network dynamics and data properties, which they claim ensures convergence to critical points in smooth non-convex optimization. Experimental results on classification tasks demonstrate that the proposed step size drives faster convergence compared to traditional learning rate schedules.

Despite some discussions, reviewers still have significant concerns regarding this work. Firstly, multiple reviewers expressed doubts about the practical relevance of assuming that the minimum loss value is known, as this is generally unrealistic in real-world applications. There were also technical concerns regarding the clarity and soundness of the main theorems, with reviewers finding the assumptions and notation inconsistent and challenging to interpret. Additionally, the experimental validation was critiqued for relying on small datasets, limiting the generalizability of the results. These issues, especially the shortcomings in the theoretical analysis, are challenging to address adequately.

**Audience:**

The problem addressed in this paper is indeed significant; however, the current results are still somewhat incomplete, and the reviewers have expressed dissatisfaction with these findings.

**Claims And Evidence:**

The theoretical analysis in this paper relies on strong assumptions, which weakens the persuasiveness of the claims.